# Vacuum laser acceleration of super-ponderomotive electrons using relativistic transparency injection

P. K. Singh[1,4], F.-Y. Li [1,4], C.-K. Huang [1], A. Moreau [2], R. Hollinger[2], A. Junghans [1], A. Favalli[1], C. Calvi[3], S. Wang[2], Y. Wang[2], H. Song [2], J. J. Rocca[2,3], R. E. Reinovsky[1] & S. Palaniyappan [1✉]

Intense lasers can accelerate electrons to very high energy over a short distance. Such compact accelerators have several potential applications including fast ignition, high energy physics, and radiography. Among the various schemes of laser-based electron acceleration, vacuum laser acceleration has the merits of super-high acceleration gradient and great simplicity. Yet its realization has been difficult because injecting free electrons into the fast-oscillating laser field is not trivial. Here we demonstrate free-electron injection and subsequent vacuum laser acceleration of electrons up to 20 MeV using the relativistic transparency effect. When a high-contrast intense laser drives a thin solid foil, electrons from the dense opaque plasma are first accelerated to near-light speed by the standing laser wave in front of the solid foil and subsequently injected into the transmitted laser field as the opaque plasma becomes relativistically transparent. It is possible to further optimize the electron injection/acceleration by manipulating the laser polarization, incident angle, and temporal pulse shaping. Our result also sheds light on the fundamental relativistic transparency process, crucial for producing secondary particle and light sources.

[1] Los Alamos National Laboratory, Los Alamos, NM 87545, USA. [2] Department of Electrical and Computer Engineering, Colorado State University, Fort Collins, CO 80523, USA. [3] Department of Physics, Colorado State University, Fort Collins, CO 80523, USA. [4] These authors contributed equally: P. K. Singh, F.-Y. Li. ✉email: sasi@lanl.gov

Table-top petawatt-class lasers provide a large electric field (>10 TV/m) capable of accelerating electrons to near-light speed over a very short distance[1–5]. Such compact accelerators have several potential applications including fast ignition, high energy physics, radiography, and secondary ion/neutron sources[6–12]. Existing schemes of laser-based electron acceleration fall into two broad categories: (1) Laser Wakefield Acceleration (LWFA)[13–17] that uses the plasma wakefield (~10 GV/m) driven by the laser to accelerate the electrons and (2) Direct or Vacuum Laser Acceleration (DLA or VLA)[18–28] where the injected electrons are directly accelerated by the intense laser field (>10 TV/m). Both the VLA and DLA schemes are similar except that in DLA a small plasma field assists the electron acceleration by reducing the electron de-phasing.

In VLA, the laser electric field initially drives the electrons in the transverse direction, and subsequently the Lorentz force ($\mathbf{v} \times \mathbf{B}$) quickly accelerates the electron in the forward direction along the laser beam. VLA normally occurs in free space, where the plasma field effect is either minimized or completely ruled out. The simplicity of the VLA scheme has attracted fundamental interest and extensive studies in the last few decades[21,29–32]. However, the grand challenge of VLA lies in how to properly load free electrons into the fast-varying laser field such that the injected electron remains within a given half cycle of the laser wave and sees a unipolar field for continuous acceleration. This requirement necessitates the injected electron to be pre-accelerated close to the speed of light (i.e., the laser speed) before it can be captured and accelerated by the intense laser field. Several schemes have been proposed to facilitate electron injection in VLA, for example, by tailoring the laser profile, ionizing highly-charged ions, or using nanocluster targets[33–35]. Nevertheless, clear experimental demonstration of VLA has been difficult[19,36,37]. Despite using relativistically intense lasers, many experiments demonstrated only 100 s keV acceleration[19]. While the scheme of ionizing highly-charged ions predicted GeV acceleration numerically[34], preliminary experiments only showed ~1 MeV photoelectrons generated from this process[36]. The scheme exploiting partially transmitted laser for acceleration between two foils showed no amplification in the energy but only an increase of electron number around 1 MeV energy[37]. Recently, a breakthrough was made in VLA using a plasma mirror injector accelerating electrons to relativistic energies around 10 MeV[25]. This injector essentially peels the surface electrons off a thick solid plasma mirror upon oblique laser incidence and then injects them into the specularly reflected laser field.

Here we demonstrate VLA of electrons up to 20 MeV using a qualitatively different injection method that exploits the plasma relativistic transparency (RT) effect[38]––where dense opaque plasma becomes transparent to the driving laser due to relativistic electron mass increase––by driving a thin solid foil at normal laser incidence. For thicker foils that remain opaque to the entire duration of the incident laser pulse, the laser-plasma interaction is constrained to the front plasma surface, resulting in the well-known ponderomotive scaling of electron energy[39], $\langle \gamma_h \rangle = \sqrt{1 + a_0^2/2}$, where $\langle \gamma_h \rangle$ is the average Lorentz factor of hot electrons and $a_0$ is the laser strength parameter. In contrast, when the laser interacts with a thin foil, super-ponderomotive electrons can be generated from VLA by using this RT effect as the injector. Our experimental results show 20 MeV super-ponderomotive electrons with four times higher flux from thin plastic foils (5 nm thick) undergoing RT injection and subsequent VLA compared to only 10 MeV electrons from thicker foils (thickness 20 nm and above) that remain opaque for the entire laser pulse duration. Therefore, our work solves an outstanding problem in VLA by demonstrating a viable injection method. Compared with the plasma mirror scheme based on the surface interaction, the present transmission scheme involves the volumetric RT process which may imply faster scaling of laser-to-electron coupling efficiency when driven by lasers with larger focal volume and has the potential to generate micro-Coulomb electron beams if scaled to picosecond laser drivers[40]. More importantly, as we shall see, our results provide insights into the electron dynamics in RT plasmas. Due to its volumetric interaction nature, the RT regime has been widely recognized as a compelling platform for laser ion accelerators, x-ray sources, and relativistic optics[8,9,11,12]. However, most previous studies have focused on these secondary sources while leaving their primary driver––fast electrons––least understood. By advancing the understanding in laser-electron coupling, our work should stimulate developments in the various secondary sources.

## Results

**Relativistic transparency injection**. The phenomenon of RT occurs when the plasma electrons are heated to near-light speed, increased in their mass, and eventually become unable to shield the plasma quickly from the driving laser[38]. It is nominally achieved as the effective plasma density, $n_e/\langle \gamma \rangle$, drops below the plasma critical density ($n_c = 1.1 \times 10^{21}/\lambda_0^2$ cm$^{-3}$ with $\lambda_0$ being the laser wavelength in microns), transforming a classically overdense plasma to be relativistically underdense. In the present work, we identify that the RT effect plays a central role in regulating the VLA as schematically shown in Fig. 1a–c. When a high-contrast intense laser is incident on a thin solid foil, it first drives surface electrons from the foil in the laser direction due to the ponderomotive $\mathbf{J} \times \mathbf{B}$ heating[41]. The resulting charge separation establishes a sheath field at the rear side of the foil, which acts as a barrier to the electrons and sends them back towards the incoming laser (Fig. 1a). Meanwhile, as the dense foil plasma ($n_e \gg n_c$) reflects the incident laser, a standing wave is formed in the front side of the plasma, where the refluxing electrons undergo a violent stochastic acceleration (discussed in detail later) and are again sent towards the target rear side with a large forward momentum. As the interaction advances, the plasma density ($n_e$) continues to decrease due to plasma expansion and the electron Lorentz factor $\langle \gamma \rangle$ increases. Near the peak of the laser pulse, the condition for the onset of the RT, $n_e/\langle \gamma \rangle \sim n_c$, is met (Fig. 1b). The onset of RT both allows the incident laser to pass through the otherwise overdense plasma and reduces the strength of the sheath field. As such, the stochastically accelerated electrons can easily escape the sheath field barrier and be injected into the transmitted laser field where they undergo phase-stable VLA until leaving the laser pulse radially via ponderomotive scattering (Fig. 1c).

## Experimental results

A schematic of the experimental setup is shown in Fig. 1d and the details are described in the Methods. Accessing the RT regime requires both a high-contrast laser pulse to avoid premature target expansion and a relatively lower plasma density than a solid foil to more easily satisfy the RT turn-on condition ($n_e/\langle \gamma \rangle < n_c$). Here, we use a unique frequency-doubled ($\lambda_0 = 400$ nm) intense laser pulse from the ALEPH petawatt laser facility at the Colorado State University (CSU)[42] to access the RT regime. The frequency-doubling both sharply increases the laser contrast and decreases the effective plasma density seen by the laser; a plastic foil has an initial target density of $280n_c$ at $\lambda_0 = 800$ nm whereas it is only $70n_c$ at 400 nm ($4 \times$ lower).

To verify the high-contrast interaction, the back-reflected laser light is monitored (Fig. 1d and also Supplementary Fig. 2). The associated spectra are indicative of the dynamics of the plasma critical surface before the onset of RT, as they encode the motion of

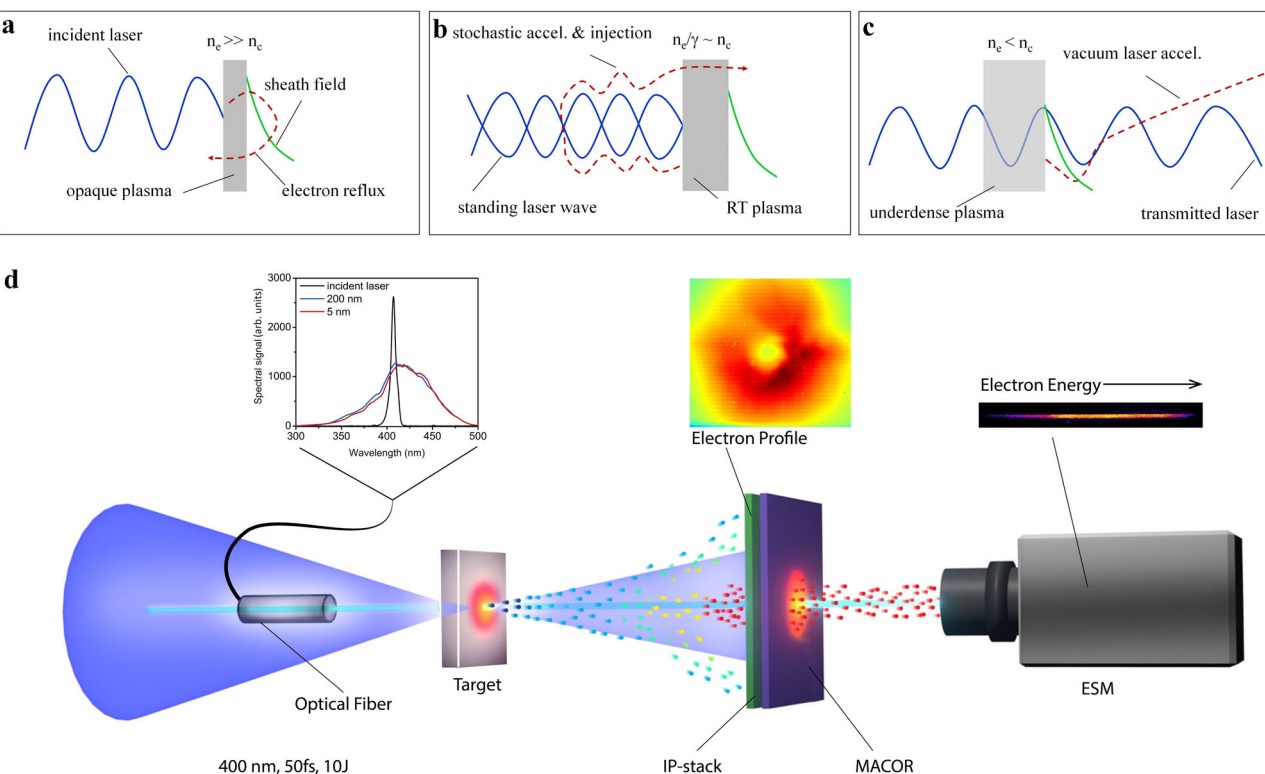

**Fig. 1 Schematic of the vacuum laser acceleration using relativistic transparency injection and the experimental setup.** (**a–c**) and the experimental setup (**d**). **a** An intense laser drives a sheath field at the rear side of the target that refluxes the electrons towards the incident laser. **b** The refluxed electrons undergo stochastic acceleration from the laser standing wave in front of the target and get injected again towards the target rear side. **c** Injected electrons undergo vacuum laser acceleration in the laser field that passes through the target due to relativistic transparency. **d** An ultra-thin nanometer target is irradiated by a 400-nm-wavelength, high-contrast, high-intensity laser pulse at normal incidence. An optical fiber, positioned at the laser beam center, collects a small portion of the light reflected from the target. The transmitted laser beam is captured by a calibrated MACOR sheet used as a calorimeter. The electrons, shown by tiny spheres, gain energy while co-propagating with the transmitted laser pulse, as indicated by a change of their color from blue to red. During propagation, the low-energy electrons are deflected away from the central laser axis by the ponderomotive force of the transmitted laser pulse. An electron spectrometer measures the kinetic energy spectrum of the on-axis electrons as they pass through a hole in the MACOR sheet. The spatial profile of the electron beam is captured by replacing MACOR with a stack of image-plates (IP-stack).

the critical surface relative to the driving laser. Initially, the plasma expands toward the incident laser causing a blue-shift to the reflected light. As the laser intensity increases and the plasma density decreases, the plasma critical surface is driven away from the laser imprinting a red-shift[38]. Overall, the spectra are significantly broadened as confirmed in Fig. 1d. Most prominently, little difference is found in the spectra by varying the foil thickness. This result, as a coarse indicator, could imply negligible pre-pulse effects and the interaction of the main pulse with a high-density plasma even for the thinnest 5 nm target.

The fast electron spectra from the opaque foils (thickness of 20–200 nm) show a similar temperature of 1.8 ± 0.4 MeV and cut-off energy of 10–12 MeV (the gray region in Fig. 2a). In contrast, the partially transparent 5 nm foils (transmitted laser energy 10–20% with the spread induced by thickness uncertainty of the thinnest targets) result in much hotter fast electrons up to 20 MeV with an electron temperature of 6.1 ± 0.5 MeV. A similar contrast also appears in the transverse beam profile of the accelerated electrons. The 200 nm opaque foil produces a more uniform electron beam profile (Fig. 2d) with a 20-degree FWHM and a slight elongation along the laser polarization axis (Fig. 2b), whereas a distinct doughnut-shaped beam profile is observed for the 5 nm thick foil (Fig. 2c, e). The FWHM of the doughnut hole is about 30 degrees (white circle in Fig. 2c and lineout in Fig. 2e), consistent with the F/2 off-axis-parabola laser beam size. This coincidence indicates that the doughnut-shaped beam profile may

be related to ponderomotive scattering from the transmitted laser pulse.

Using simultaneous measurements of the transmitted laser energy and electron spectrum, we obtain a direct correlation between the fast electron heating and the amount of laser transmission. At 10–20% transmission, the maximum electron energy quickly rises to ~22 MeV, a factor of two higher than from the opaque targets (Fig. 3a). A similar trend is also observed for the integrated electron flux in the 6–20 MeV range (Fig. 3b), which increases by a factor of four as compared to the opaque regime.

**Modeling of the experimental results.** To understand the dynamics of electron acceleration in the RT regime, we perform two-dimensional (2D) particle-in-cell (PIC) simulations using the SMILEI code[43]. The simulations are mainly used to elucidate the key physics that underpin the significantly enhanced acceleration observed in the RT regime. Quantitatively reproducing all observables using the nominal laser-target conditions is subject to the uncertainties in the target fabrication, vapor contamination, and on-target laser intensity and focal spot quality. Therefore, we have explored a range of foil thickness and laser intensity closely around the nominal parameters, and these simulations show qualitatively the same physics of electron acceleration (detailed below) for 5~30 nm foils once the RT sets in, despite varying amounts of laser transmission and electron acceleration

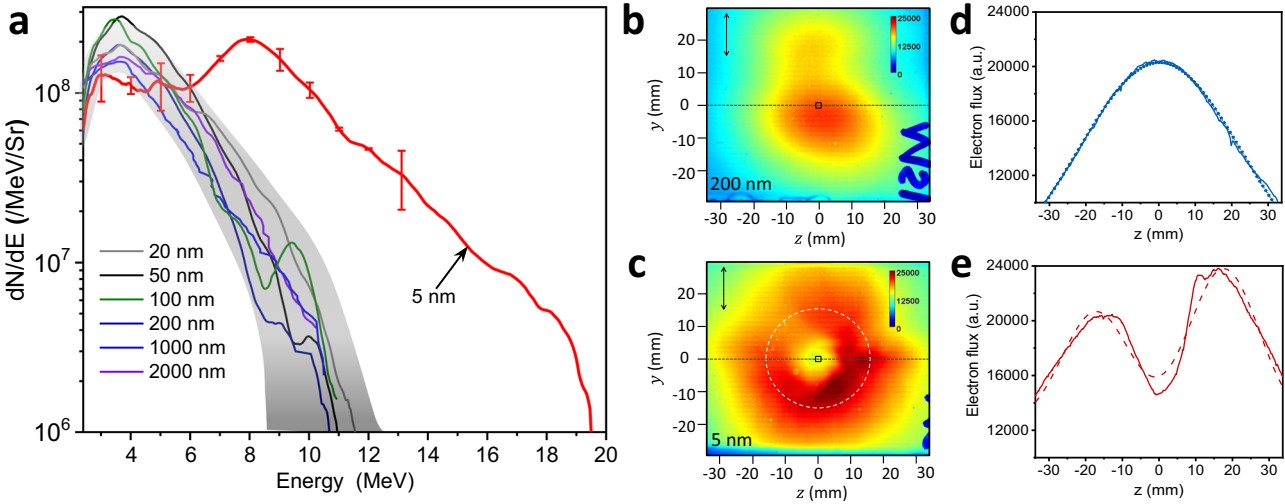

**Fig. 2 Spectral and spatial influence of transparency on MeV electrons. a** The kinetic energy spectrum of electrons is recorded on-axis for different target thickness. The gray shaded region shows the spectral bounds from all opaque targets (20–2000 nm thick). The error bars represent the standard deviation of the data from all the corresponding shots. The spatial profile of the electron beam recorded on the third layer of the image plates (energy > 1 MeV), in a plane perpendicular to the laser propagation direction, is shown for **b** the opaque 200 nm and **c** partially transparent 5 nm targets. The black squares in the center of **b** and **c** represent the acceptance window of the electron spectrometer, and the vertical arrows indicate the laser polarization direction. The white dotted circle in **c** represents the size of the transmitted laser beam. **d**, **e** show lineouts of corresponding electron beam profiles taken along the black horizontal lines in **b** and **c**. The dashed curves in **d** and **e** represent Gaussian fits to the solid curves.

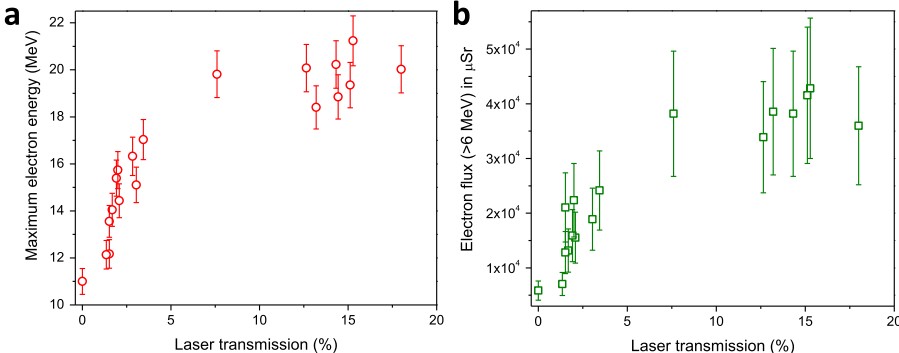

**Fig. 3 Transparency dependent electron acceleration. a** Maximum electron energy and **b** total electron flux per μSr for energy greater than 6 MeV as a function of laser-transmission ratio. The error bars represent the standard deviation of the data from all the corresponding shots.

(Supplementary Figs. 4 and 5). Without loss of generality, we take the 20 nm and 200 nm case as representative of the RT and opaque regime, respectively, and focus on identifying the acceleration mechanism pertinent to the RT regime. In both cases, the electron density ($n_e$) first rises due to the radiation pressure compression[44,45] followed by a continuous drop as the foil undergoes expansion (Fig. 4a). Comparing to the effective density ($n_e/\langle\gamma\rangle$), however, the thicker 200 nm case shows only marginal drop, indicating limited bulk electron heating. This latter point is illustrated in Fig. 4c by a snapshot of the cell-averaged $\langle\gamma\rangle$ at $t = 10T_0$ ($T_0$ being the laser cycle), where the electron heating is constrained to the front surface with the bulk of the foil being largely cold. For this 200 nm case, the foil stays opaque throughout the 50 fs driving laser by retaining $n_e/\langle\gamma\rangle > 20n_c$ at $t = 50T_0$, the end of the laser interaction.

In contrast, the thinner 20 nm foil nearly matches the condition, $d \leq \frac{a_0}{\pi}\frac{n_c}{n_e}\lambda_0$, for the radiation pressure acceleration[46], where $a_0$ is the laser amplitude. Consequently, the foil center including both electrons and ions is pushed more efficiently from its initial position. Direct laser heating of the resulting bowed plasma surface then leads to rapid foil expansion and a sharp density drop around $t = -15T_0$ (Fig. 4a). Most prominently, this

trend is greatly amplified by the relativistic $\langle\gamma\rangle$-correction arising from the strong heating (compare the red lines), and the RT quickly sets in as the condition $\frac{n_e}{\langle\gamma\rangle n_c} < 1$ is reached at $t = -5T_0$, shortly before the arrival of the laser peak. The plasma becomes classically underdense ($n_e/n_c < 1$) in another 30 laser cycles, during which the laser transmits through the plasma and drives a significant portion of electrons that co-propagate with the pulse (Fig. 4d). As we shall see, these electrons are responsible for the greatly extended energy spectrum (Fig. 4b) and the doughnut-shaped profile as opposed to that from the opaque regime (right panels of Fig. 4c, d). These major imprints of the distinct interaction regimes are in good agreement with the experimental observations (Fig. 2).

**Overview of electron acceleration in the relativistic transparency regime.** Next, we discuss electron acceleration in the RT regime in more detail by tracking representative electrons from the simulation. The resulting trajectories (Fig. 5a) show that the strongest acceleration (colors turn red) occurs away from the initial foil position ($x > 25\lambda_0$). This is an indication of VLA[18,25] by the transmitted laser pulse as the plasma sheath field lags due to

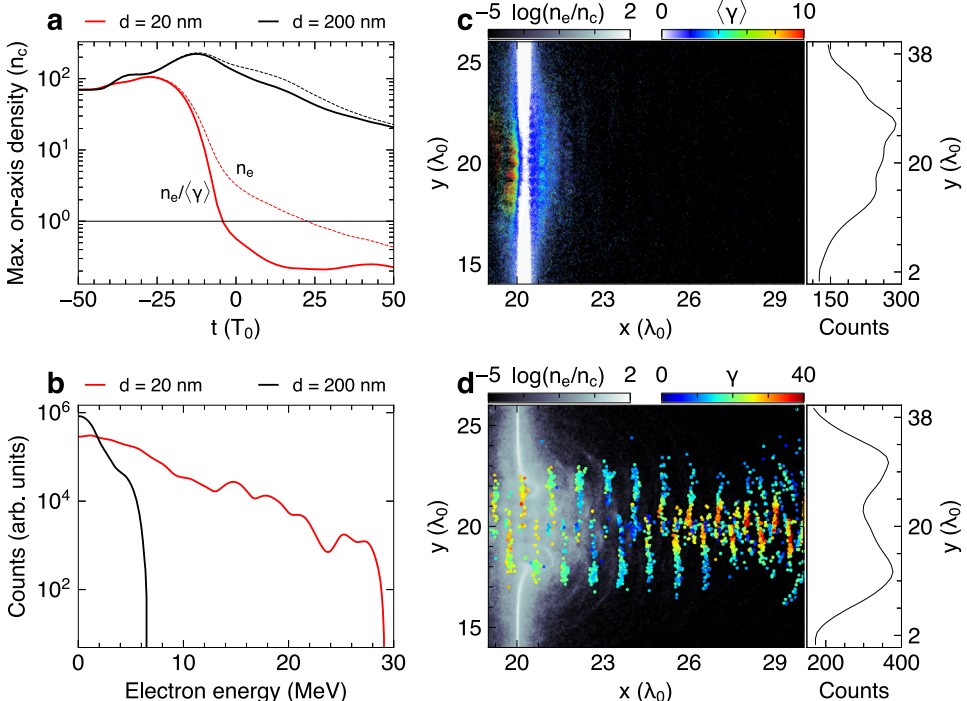

**Fig. 4 Relativistic transparency versus opaque regime with 2D PIC simulations. a** Temporal evolution of maximum on-axis electron density (both normal density $n_e$ and effective density $n_e/\langle\gamma\rangle$) for laser-driven 20 nm and 200 nm foils. The driving laser is $100T_0$ long and $t = 0$ refers to the arrival of the laser peak at the initial target position. **b** The resulting electron energy spectra at the simulation end, obtained from a virtual detector placed $35\lambda_0$ behind the foil. The acceptance width is limited to $\Delta y = 2\lambda_0$ around the central laser axis. **c**, **d** Snapshots of electron density at $t = 10T_0$ for the 200 nm and 20 nm case, respectively. Also overlaid in **c** is cell-averaged $\gamma$ of all electrons and in **d** instantaneous $\gamma$ for fast electrons acquiring a maximum $p_x/(m_e c) > 25$ during the interaction. The right panels of **c** and **d** are the beam profile obtained at the simulation end on the same detector (but without $y$ limitation) as for **b**.

the slow ion motion. The trajectories also reveal deflections from the central laser axis ($y = 20\lambda_0$), inversely correlated to their final energy gain. This phenomenon explains our on-axis (acceptance angle 16 μSr) electron spectrometer measurements which missed many of the deflected electrons in the RT regime and accordingly recorded a lower flux in the energy range below 8 MeV (Figs. 2a and 4b).

Displayed as the phasespace of $p_x$ versus $x$ (Fig. 5b), all the tracked electrons strikingly follow the same dynamics despite the spread in their final energy gain. The full dynamics may be divided into (1) a phase-space rotation around the foil (including refluxing and foil front acceleration), (2) a brief deceleration as electrons escape the rear sheath barrier, and (3) a final injection and acceleration in the transmitted laser pulse, as labeled in Fig. 5b. When the laser interacts with the foil, the $\mathbf{J} \times \mathbf{B}$ heating[41] first drives electrons into the target. Some of them may return to the front side after bouncing off the resulting sheath (charge-separation) field. The refluxed electrons get accelerated again in front of the foil (detailed below), thus closing the loop of the phase-space rotation. Before the onset of RT, electrons may undergo several cycles of rotation but gain no further acceleration in the rear side with the transmitted laser field yet to appear. Therefore, the presented cycle of rotation leading to the enhanced acceleration must be near the onset of RT. By checking the temporal information of a typical electron (blue line in Fig. 5b) as shown in Fig. 5c, the reflux is indeed initiated shortly before the laser starts to transmit through ($t = -5T_0$ as explained in Fig. 4a).

**Stochastic electron acceleration in front of the foil and relativistic transparency injection**. The ensuing sharp reversal of $p_x$ in front of the foil (Fig. 5b) involves electron interaction with a complicated field configuration including both the laser fields and

self-generated plasma fields. To see what causes the front-side acceleration, we perform simplified test-particle simulations (see the Methods for detailed setup) where the above complicated fields can be separated easily and added back one by one. These simulations reveal that the acceleration is mainly caused by the interfering incident and reflected laser pulses, and the energy gain depends critically on the laser amplitude, the injection momentum, and injection time/phase, while being less dependent on detailed laser spatiotemporal profile, the self-generated plasma fields, and the minor spectral shifts of the reflected pulse. More specifically, the electron phase-space dynamics evolve around the closed orbits (blue buckets in Fig. 6a) arising from the ponderomotive force of the standing wave. Such buckets mapped out by actual electron trajectories can be clearly seen in both the test-particle (Fig. 6b) and PIC results (inset of Fig. 5b). For injection $p_{x0}$ smaller than the bucket height (magenta line in Fig. 6a), the electrons leave the standing wave after a half-cycle rotation due to the blocking foil. This type of dynamics is responsible for the normal $\mathbf{J} \times \mathbf{B}$ heating, generating electron pulses at every half cycle. For $p_{x0}$ beyond the bucket height (red/green lines), the electrons can penetrate deeper into the left. Some of them (red) may be temporarily trapped away from the foil, whereas others (green) continue to wander around the buckets, resulting in the sharp acceleration at the foil front. For even larger $p_{x0}$ (cyan), the electrons may not be trapped by any buckets but simply form backward ejections.

To further clarify the acceleration, we transform these dynamics into a Poincare map (see the Methods for details) which represents the traces of the electrons on a phasespace cross-section that is commonly used to distinguish the stability (i.e., stochastic or regular motion) of their quasi-periodic orbits. Two sets of electrons are initialized as shown in Fig. 6c. The electrons

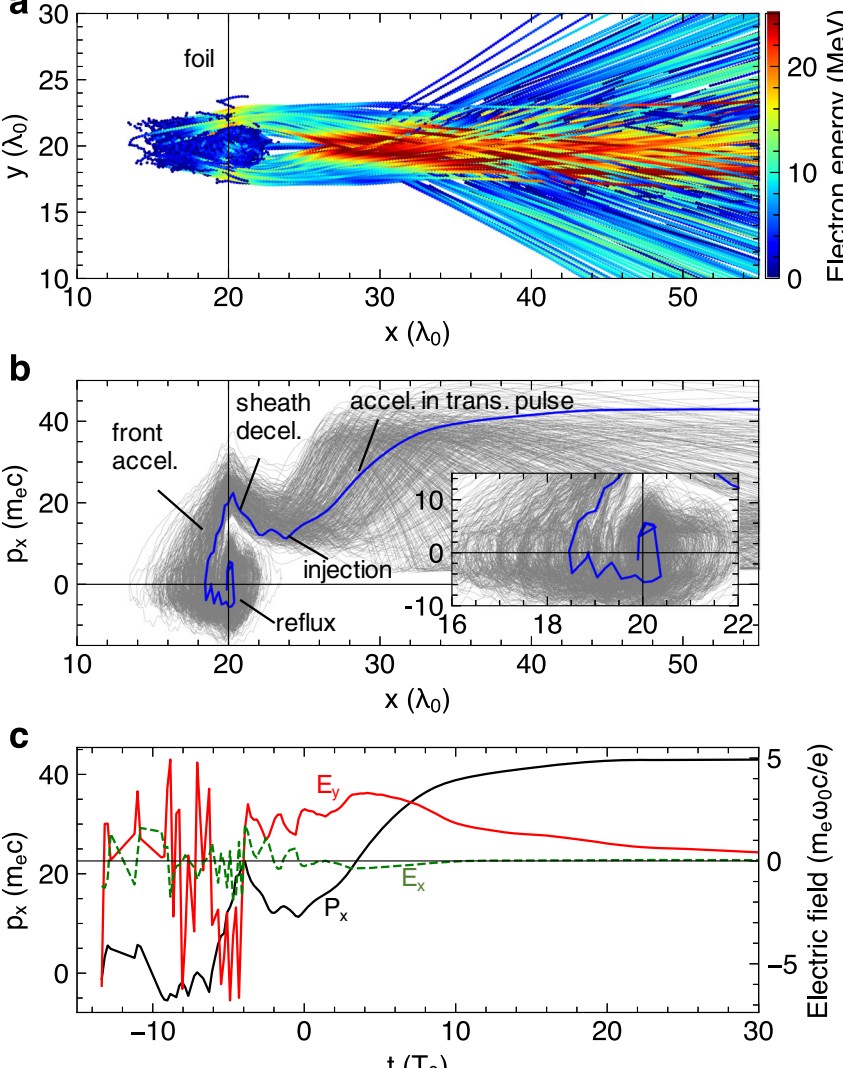

**Fig. 5 Multi-stage electron acceleration in the relativistic transparency regime. a** Trajectories of the electrons that acquire a maximum $p_x/(m_e c) > 35$ in laser-driven 20 nm foil, with colors representing instantaneous energy variation. **b** Corresponding phase-space ($p_x$ – $x$) trajectories with the inset providing a closer look around the initial foil position ($x = 20\lambda_0$). **c** Temporal evolution of $p_x$ and experienced $E_x$, $E_y$ fields for a selected electron (blue line in **b**). We have also checked with a lower threshold $p_x^{max}/m_e c > 25$ (e.g., in Fig. 4d), which results in similar dynamics except for greatly more particles. This latter condition almost covers the entire extended-spectrum (Fig. 4b), thus the presented dynamics is representative of the observed fast electrons.

with smaller $p_{x0}$ (red dots) become quickly dispersed after just one laser cycle, indicating the onset of stochasticity. That is, these electrons may be briefly in phase with either of the lasers due to perturbation by the other laser, and be stochastically accelerated to any point of the area in a few laser cycles. In contrast, the electrons with much larger $p_{x0}$ (black dots) show only regular traces underneath the stochastic area with $p_x$ being kept negative. Therefore, the sharp $p_x$ reversal to positive values as found in the simulations must be caused by the stochastic acceleration[47]. At relativistic laser intensities, even the normal $\mathbf{J} \times \mathbf{B}$ heating could be stochastic if it were not terminated by the blocking foil. The phase dependence of the acceleration can be illustrated by the blue dots (Fig. 6c) with a $\pi/2$ shift in the injection phase; despite having similar $p_{x0}$ with the red dots, they share the same regular motion as the black dots (thus not shown). We have found numerically that the maximum $p_x$ achievable via the stochastic acceleration scales linearly with the laser amplitude (Fig. 6d), giving $p_x/(m_e c)$ ~20 at $a = 5$. This result accounts for about 75% of the front-side acceleration as observed in the PIC simulation, with the rest of the acceleration possibly assisted by the self-generated fields near the

bulk of the target (see Fig. 5c for the relative weights of $E_x$ and $E_y$ during $t = [-10, -4]T_0$). Moreover, the short timescale involved suggests that the effective laser amplitude for stochastic acceleration can be mapped to the timing of the onset of RT (Fig. 6d). For example, if the transparency occurred too early/late (right vertical axis), the corresponding laser amplitude (horizontal axis) and the maximum $p_x$ from the stochastic acceleration (left vertical axis) would be small. Thus, it predicts strongest front-side acceleration if the RT is induced near $t = 0$ where the laser amplitude peaks.

**Vacuum laser acceleration in the transmitted laser pulse.** The sharp front-side acceleration ensures that the electrons retain a positive $p_x$ after briefly traversing the plasma-sheath barrier in the rear side. Indeed, nearly all of the tracked electrons in the PIC simulation have minimum $p_x > 10$ $m_e c$ at x~23$\lambda_0$ (Fig. 5b). The residual $p_x$ favors subsequent injection into the transmitted laser pulse by reducing the electron-laser dephasing rate[48]. As a result, the electrons experience a slow-varying laser field $E_y$ in the rear space of vanished plasma field $E_x$ (red/green lines in Fig. 5c at $t > 0$). Such phase-locked acceleration of all tracked electrons is

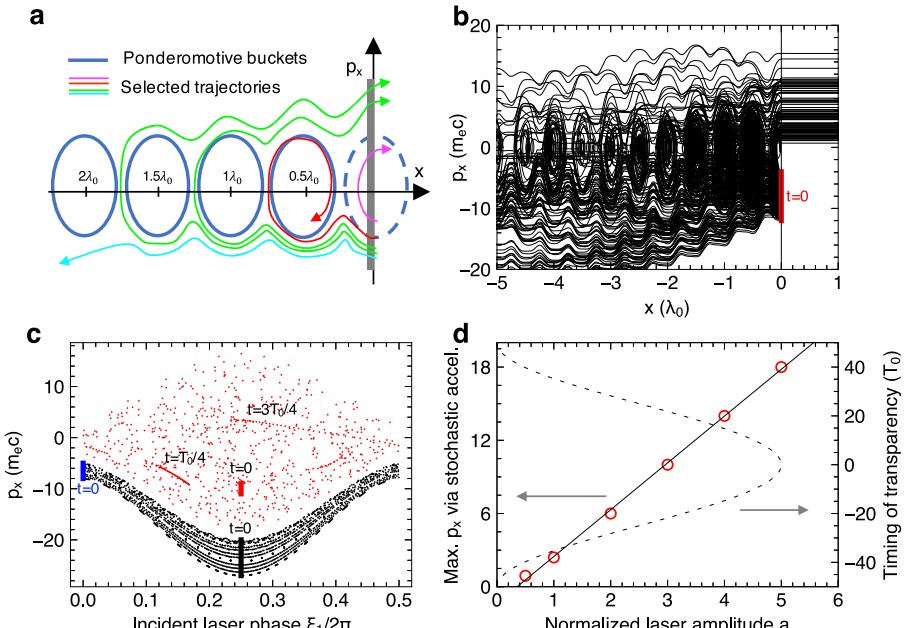

**Fig. 6 Stochastic acceleration in front of the foil identified with test-particle simulation and Poincare-map analysis. a** Schematic of typical phase-space trajectories of the refluxing electrons ($p_x < 0$) that are injected into the standing laser wave formed in front of the foil. **b** Similar trajectories were obtained from a test-particle simulation. A group of 200 electrons is injected at $x = 0$ with uniformly distributed $p_x0 = [-12, -4]$ $m_e c$ (red line). The injection time is randomly sampled over the first $T_0/2$. To mimic the laser fields close to the transparency time, the laser wave takes a time delay of $t_d = -40T_0$. **c** Poincare map of the same electron dynamics ($p_x$ versus $\xi_1$) displayed periodically at the surface of the section $\xi = \xi_1 + \xi_2 = 2N\pi$ where $\xi_1$ and $\xi_2$ are the phases of the incident and reflected laser, respectively. **d** Maximum $p_x$ achievable via the stochastic acceleration (red circles) versus laser amplitude with the latter being also mapped to the assumed onset timing of relativistic transparency. The solid line is a linear fitting to the circles. See the Methods for more details of the test-particle simulation setup and analysis.

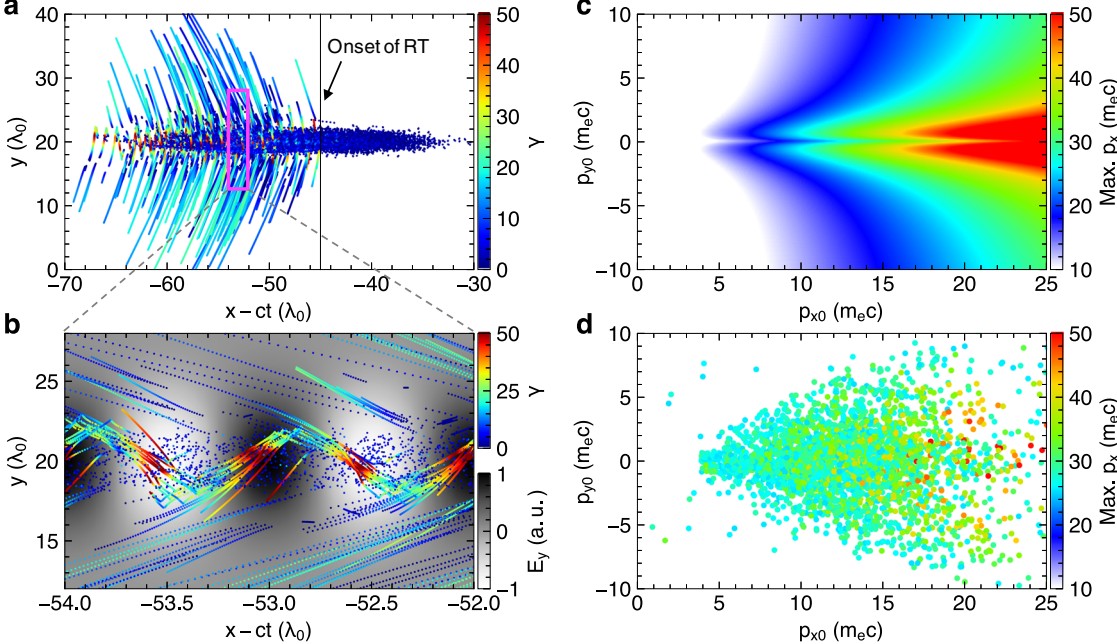

**Fig. 7 Electron acceleration in the transmitted laser pulse. a** Trajectories of the same set of electrons as shown in Fig. 5 but displayed now in the co-moving frame of the laser, x-ct (Fig. 7a). **b** Zoom-in of the rectangular area marked out in **a**. The background shows the laser electric field. **c** Analytical maximum $p_x$ achieved in vacuum plane-wave laser acceleration versus initial injection momenta ($p_x0$, $p_y0$) for normalized laser amplitude of $a_0 = 1$. **d** The same type of distribution as **c**, but obtained for the tracked particles shown in **a**.

revealed by plotting electron trajectories in the co-moving laser frame, x-ct (Fig. 7a). Electrons stream from right to left in this frame and substantial energy gain only happens after the onset of RT (vertical line). Importantly, the accelerations complete in just

half laser wavelengths (Fig. 7b) despite tens of wavelengths propagation in the lab frame (Fig. 5a). The ponderomotive deflection angle is related to the specific laser phase into which the electrons are injected and overall displays up-down symmetry by

integrating over many laser cycles (see also Fig. 4d). Also depending on the injection condition, the acceleration spreads widely over the latter half of the laser pulse (x-ct < −50$T_0$). Nevertheless, more quantitative insight into the vacuum laser acceleration[18,25] can be obtained by relating the energy gain to their injection momenta. Figure 7c shows the analytical result for a plane-wave laser of $a_0 = 1$, where the smaller laser amplitude (than incident laser $a_0 = 5$) is used to account for the rapid laser diffraction as well as the ponderomotive deflection that prevents electrons from fully exploiting the laser fields. As a comparison, the corresponding simulation result by taking ($p_{x0}$, $p_{y0}$) at $x = 23\lambda_0$ (Fig. 7d) shows good agreement with Fig. 7c, not only in the pattern of $p_x$ gains but also in the peculiar triangular distribution of ($p_{x0}$, $p_{y0}$) that favors strong acceleration.

**Validation using 3D PIC simulation**. To further validate our major findings and check possible 3D effects, we have performed 3D PIC simulations using the same set of laser-foil parameters (see Fig. 8 and the Supplementary Information). In agreement with the 2D results, the 3D simulations display the same sharp contrast as found in the electron beam profile (Fig. 8b, c) and energy spectrum (Fig. 8e) between the case of an opaque foil ($d = 200$ nm) and a RT foil ($d = 20$ nm). In the RT case the overall interaction dynamics depicted by the density evolution (Fig. 8d) also agree fairly well with the 2D result. The 3D particle tracking (Fig. 8a) again confirms very similar acceleration dynamics with the electrons first revolving around the initial target and then acquiring the most acceleration by the transmitted laser in the rear space.

These 3D simulations thus demonstrate the validity of the acceleration physics identified in this work. Minor differences from the 2D interaction also appear. Firstly, 3D simulation generally shows more target expansion than 2D simulation (Fig. 8d), partially due to the stronger laser self-focusing in 3D (Supplementary Fig. 6) and also by slight density exaggeration in 2D. Nevertheless, 3D is found to give a very close onset time of RT in this case, likely caused by the exaggerated target heating in 2D P-polarized simulations. Secondly, 3D has more rapid laser diffraction than 2D (Supplementary Fig. 6). As such, the 3D electron energy cutoff obtained in the RT regime is slightly lower (comparing Fig. 8e and Fig. 4b) due to the smaller laser amplitudes available for acceleration in the rear space. These differences are, however, quantitative and do not alter our understanding of the RT-based VLA physics. Mitigating the differences and unwanted effects will be of interest for future optimizations.

## Discussion
In summary, we have presented an elegant method to inject near-light speed electrons into a relativistic laser pulse and demonstrated clear evidence of VLA using the RT effect of laser-foil interaction. This method results in 10 s MeV super-ponderomotive electrons, and the energy could be much higher by scaling to higher laser intensities. The scaling is expected to work because the RT effect is ubiquitous as long as the laser-plasma conditions are properly matched[12,38] and the VLA bears almost linear dependence with the laser intensity[18,33]. It thus implies great flexibility in optimization of the RT-injection based VLA by manipulating the laser

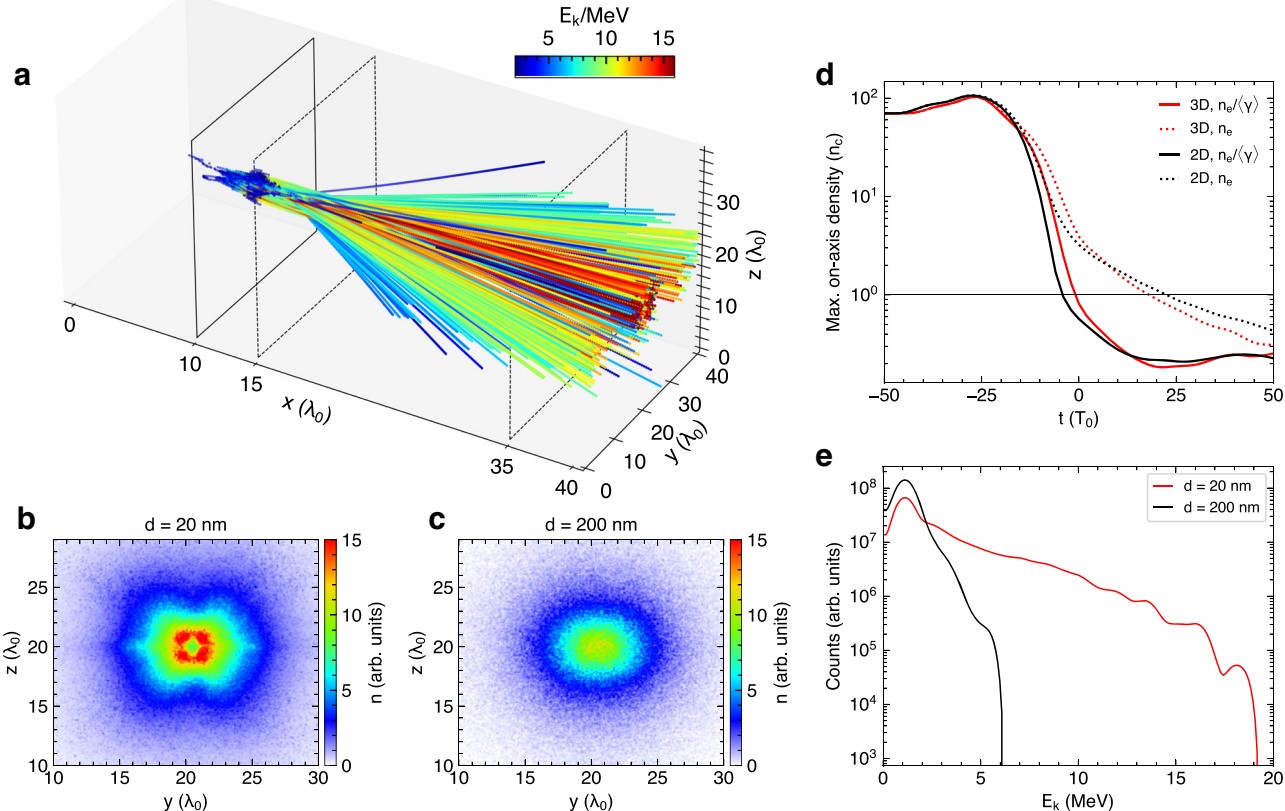

**Fig. 8 Validation with 3D PIC simulation. a** Trajectories (colored by energy) of the electrons acquiring a maximum momentum $p_x/(m_e c) > 25$ during laser interaction with a 20 nm foil. The foil initially sits at $x = 10\lambda_0$, marked by the black solid lines. **b, c** Electron beam profiles (color representing beam density) received by a virtual screen ($x = 15\lambda_0$) for the interaction with a 20, 200 nm foil, respectively. **d** Evolution of maximum on-axis density (averaged over a radius of $0.5\lambda_0$). The 2D results (black lines) are also appended for comparison. **e** Electron energy spectrum collected by a virtual screen ($x = 35\lambda_0$) for the interaction with 20, 200 nm foil, respectively.

polarization, incident angle, and spatiotemporal profile, as well as the plasma conditions. We hope the present work will stimulate considerable interest in VLA demonstration and optimization. Finally, the detailed dynamics of fast electrons uncovered in this work offers a unique viewpoint of the RT process which is normally difficult to probe via conventional optical methods[38]. These understandings will be crucial for further development in intense laser interaction with thin foils and associated secondary particle and photon sources.

## Methods

**Experimental setup.** The experiment was performed using the PW laser at the Advanced Beam Laboratory of CSU. The fundamental ($\lambda_0 = 800$ nm) laser beam is frequency-doubled in a 0.8 mm thick type-1 KDP crystal, which drastically improves the temporal contrast from $5 \times 10^{-6}$ to $10^{-12}$ at $-25$ ps. The resulting 400 nm pulse of 75 fs duration, 6.8 J maximum energy, is separated from the fundamental beam by using a sequence of five dichroic mirrors (99.9% reflective @ 400 nm and 99.5% transmissive @ 800 nm) and then focused by an f/2 off-axis-parabolic (OAP) dielectric mirror onto the target at near-normal incidence (Fig. 1). The OAP focal spot is measured, both at low and full laser power levels, to be less than 2 μm using a magnified long-working-distance objective lens coupled to a CCD camera; details are provided in the Supplementary Information. Before each shot, the target is imaged using the same objective camera to ensure precise positioning at the laser focal plane. The targets are mainly Formvar films (5 nm to 2000-nm thick), purchased from Electron Microscopy Science (EMS). A small fraction of the laser light back-reflected from the target is captured by an optical fiber sitting in the central part of the laser beam. The optical fiber is tapered to minimize degradation in the focal spot quality of the laser. The transmitted laser pulse is monitored by an imaging camera looking at a Ceramic (MACOR) sheet kept along the laser axis. The imaging system is calibrated by firing the laser pulse in a vacuum without targets. The MACOR calibration is found to be linear for laser energy up to 3 J. A hole of 7 mm diameter is made at the center of the MACOR sheet to perform simultaneous spectral measurement of on-axis electrons using a magnetic spectrometer. The spectrometer positioned at a distance of 50 cm from the target has a 2 mm × mm sized aperture (made of 12.7 mm thick stainless steel) and hence a collection angle of about 16 μSr. The angular distribution of all fast electrons including those deflected beyond the hole/spectrometer is measured separately using a multi-layer image plate (BAS-IP MS), positioned 56 mm away from the target and shielded with a 220 μm thick Cu foil to block low-energy x-rays, ions, and electrons.

**Particle-in-cell simulations.** The 2D SMILEI simulations have a box dimension of $60\lambda_0 \times 40\lambda_0$ divided into $12,000 \times 1600$ cells along the laser propagation axis $x$ and transverse axis $y$, respectively. The foil consisted of fully ionized carbon ions ($10n_c$), protons ($10n_c$) and electrons ($70n_c$) is initially placed at $x = 20\lambda_0$ with the ion species sampled by 100 macro-particles per cell (ppc) and electron species by 400 ppc. Fourth-order interpolation and isotropic temperatures of 10 eV are applied to all species. A p-polarized laser (electric field along y) is launched from the left box edge with a peak intensity $2.14 \times 10^{20}$ W/cm$^2$, corresponding to $a_0 = 5$ for $\lambda_0 = 400$ nm. A supplementary simulation (Supplementary Fig. 5) using the nominal laser amplitude $a_0 = 10$ (corresponding to peak intensity $0.86 \times 10^{21}$ W/cm$^2$) shows a similar contrast between the RT and opaque regime (Supplementary Fig. 5a, b) and similar VLA physics (Supplementary Fig. 5c) despite higher electron energy obtained with higher $a_0$. The laser field takes sine-squared temporal profile with full duration $100T_0$ and Gaussian transverse profile of spot radius $3\lambda_0$. A virtual detector placed $35\lambda_0$ behind the foil is used to collect accelerated particles. The convergence of the simulations is tested against doubling the spatiotemporal resolution as well as the total particle number. Full 3D simulations for both the 20 nm and 200 nm foil cases are also performed where the third box dimension is $38\lambda_0$. The high resolution needed in x direction to resolve the extremely thin targets and the large rear space needed for VLA to sufficiently develop make these 3D simulations very challenging. As such, we halve the transverse resolution to 20 cells per wavelength (while maintaining 200 cells per wavelength for the longitudinal resolution), and further divide the electron species into two clumps; for the central clump within the radius $R \le 5\lambda_0$ (directly under laser impinging) we use 343 ppc and otherwise we use 8 ppc, the same as for both the ion species, where $R = \sqrt{(y - y_c)^2 + (z - z_c)^2}$ and ($y_c$, $z_c$) are the transverse coordinates of the central laser axis. An additional virtual detector placed $5\lambda_0$ behind the foil is inserted to record the hot electrons that accompany the ion beams which do not reach the far detector for the limited simulation time.

**Test-particle simulation and Poincare-map analysis.** The simulation solves the electron dynamics by directly integrating the relativistic equations of motion: $d\mathbf{P}/dt = -\mathbf{E} - \mathbf{P}/\gamma \times \mathbf{B}, d\mathbf{r}/dt = \mathbf{P}/\gamma, d\gamma/dt = -\mathbf{E} \cdot \mathbf{P}/\gamma$, where the vectors denote the three Cartesian components and the quantities are normalized as $p \to m_e c, r \to c/\omega_0, t \to 1/\omega_0, E \to m_e \omega_0 c/e, B \to m_e \omega_0/e$. A perfect light reflector is defined at $x = 0$ to mimic the foil before the onset of RT. As a result, the laser fields in front of the foil ($x < 0$) include both the incident laser $E_{y1} = B_{z1} = -af_1(t)g(r)\cos(t - x - t_d + \phi_1)$ and the reflected laser $E_{y2} = -B_{z2} = -af_2(t)g(r)\cos[\omega_2(t + x) - t_d + \phi_2]$, where $f_1(t) = \sin^2[\pi(t - x - t_d)/\tau] \times [H_2(t - x - \tau - t_d) - H(x)], f_2(t) = \sin^2[\pi(t + x - t_d)/\tau] \times [H(t + x - t_d) - H(x)], g(r) = \exp[-(y^2 + z^2)/\sigma^2]$ define the temporal and spatial profiles, $H$ is the Heaviside step function, $H_2 = 1 - H, \omega_2$ is the angular frequency of the reflected laser normalized to that of the incident laser, and the initial phases satisfy $\phi_2 = \phi_1 + \pi$. The pulse duration $\tau$ and spot radius $\sigma$ follow that of the PIC simulation. The parameter $t_d$ introduces a time delay for the incident laser to arrive at the foil, thus $t_d < 0$ means the laser has impinged on the foil for a period of $|t_d|$. To obtain the Poincare map shown in Fig. 6c, we transform the independent variable of the above equations of motion from $t$ to $\xi = \xi_1 + \xi_2$ where $\xi_1 = t - x + \phi_1$ and $\xi_2 = \omega_2 (t + x) + \phi_2$ are the phase of the incident and reflected laser, respectively. The resulting equations of motion become $d\mathbf{P}/d\xi = (-\mathbf{E} - \mathbf{P}/\gamma \times \mathbf{B})/(d\xi/dt), d\gamma/d\xi = (-\mathbf{E} \cdot \mathbf{P}/\gamma)/(d\xi/dt), d\xi_1/d\xi = (1 - p_x/\gamma)/(d\xi/dt), d\xi_2/d\xi = \omega_2(1 + p_x/\gamma)/(d\xi/dt)$ and are solved only at periodic $\xi = 2N\pi$ where $d\xi/dt = (1 - p_x/\gamma) + \omega_2(1 + p_x/\gamma)$.

## Data availability

The datasets generated and/or analyzed during the current study are available from the corresponding author upon reasonable request.

## Code availability

SMILEI is an open-source code available at: https://github.com/SmileiPIC/Smilei. Our test-particle codes are available from the corresponding author upon reasonable request.

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

## Acknowledgements

The research was almost entirely supported by the Laboratory Directed Research and Development (LDRD) Program of Los Alamos National Laboratory (LANL) under the project 20190124ER and partially by LANL Science Campaign-3. This research used resources provided by the Los Alamos National Laboratory Institutional Computing Program, which is supported by the U.S. Department of Energy National Nuclear Security Administration under Contract No. 89233218CNA000001. The experiments were conducted at the CSU ALEPH laser facility supported by the US Department of Energy LaserNet US grant DE-SC-0019076. CSU researchers acknowledge the support of AFOSR award FA9550-17-1-0278 and DOD Vannevar Bush Faculty Fellowship ONR award N000142012842. We also thank Juan C. Fernandez for useful discussions.

## Author contributions

S.P. designed the experiment. S.P., C.K.H., and J.R. provided the overall guidance of the project. P.K.S., A.M., and R.H. carried out the experiment with support from S.W., Y.W., H.S., C.C., A.J., A.F., and R.R. F.Y.L. performed all the numerical work and clarified the acceleration mechanism with inputs from C.K.H., P.K.S., and S.P. The manuscript was prepared by P.K.S., F.Y.L., S.P., and C.K.H., and reviewed by all authors.

## Competing interests

The authors declare no competing interests.
