## [Peer Review File · Nature Communications]

Vacuum laser acceleration of super-ponderomotive electrons using relativistic transparency injectionREVIEWER COMMENTS

Reviewer #1 (Remarks to the Author):

Review

The manuscript presents an important finding on relativistic transparency (RT). Namely, dramatic electron spectra and angular distributions are presented that suggest the onset of RT, which is still a relatively unexplored phenomenon *experimentally*. The experimental results combined with PIC simulations also indicate a novel electron acceleration mechanism at play.

The experiment seems well performed and the results are striking. I have some concerns about the analysis, although I expect they can be addressed. This work is described in sufficient detail that both experiment and simulation could be reproduced. The subject area and findings are appropriate for Nat. Comm. The writing was conversational and generally straightforward to follow.

Discussion

Relativistic transparency is a key phenomenon in a wide range of intense laser-plasma experiments. It is of fundamental importance because it is an example of the interaction of light with a relativistic medium and not just isolated relativistic particles. I think this makes this area of broad interest. Surprisingly, however, there are relatively few experiments that directly measure this effect, with reliance on PIC simulations usually the primary means of identifying its presence. Although PIC plays a key role here, the data are compelling, as summarized in Fig. 2. The difference between the 20 nm and 5 nm thick film spectra is suggestive. The concomitant change in the angular distribution is also persuasive. I don't see any flaws in the performance of the experiment. It may seem surprising that such an experiment has not been performed already with ultrashort lasers (as opposed to lasers like Trident or Phelix), but few such lasers have the contrast needed (at least in the US) and the targets are harder to work with.

I disagree somewhat with the motivation provided for this work. The primary justification given seems to be development of a source of accelerated electrons. However, laser plasma accelerators (eg. wakefield) provide a far better electron beam in energy, spectrum, yield and angular distribution and with far less restrictive requirements on both the laser and target. However, I find their argument that these results provide new insight on RT persuasive and sufficient. Stochastic electron dynamics have long been identified and the importance of the target in dephasing accelerated electrons is well known (ie. JxB). However, the mechanism here depends on shaping of the electron momentum distribution in front of the target due to the transient standing wave followed by achievement of RT and transmission

of the laser through the target with injected electrons. Substantial acceleration occurs far from the target.

Concerns

The PIC simulations play a critical role in this work and, indeed, their discussion takes up most of the manuscript. I do think the primary conclusions derived from the simulations are likely correct, but I have some concerns.

1) The biggest concern is the use of 2D simulations. Although I am not personally familiar with SMILEI, the PIC code used, 3D simulations of an experiment with these parameters is not out of reach any more. (I do recognize a bigger grid is needed to see the acceleration.) Still, use of 2D here is surprising. As the authors are aware, there are several issues. 2D tends to exaggerate the density of the target as it heats and, of course, the density is critical to RT. Also, 2D can greatly exaggerate and extend the sheath field which will affect the injection of the electrons into the pulse. Finally, although I think this a small effect here, the phase and amplitude of a 2D Gaussian approaches the target differently than that for 3D. However, it is stated that a full 3D simulation was indeed performed with very similar results. It is surprising these weren't used for the analysis, then. In any case, they should be shown, preferably in a figure corresponding to Fig. 2.

2) The experiment shows a dramatic difference between the 20 nm and 5 nm targets. I think that is a primary strength of the paper, in fact. However, the discussion suggests the simulations don't see a dramatic difference (see also Fig S3) and, indeed, 20 nm is used to compare to the experimental 5 nm. This should be discussed. I'm guessing the electron spectra differed significantly between 5 nm and 20 nm in the simulations. Was the thickness used as a tuning parameter? If so, this doesn't necessarily invalidate the major claims, but we should be informed.

The remainder of my concerns are lesser and, I think, readily addressed.

A) I disagree with the simple model explaining spectral broadening (page 4, merged pdf). It makes it seem as if it is all simple Doppler shifts, whereas phase modulation due to ionization and, especially, density variation, is just as strong.

B) It is emphasized that the independence of the spectra on target thickness is clear evidence of good contrast. Why is this true? The front surface physics will be the same under these conditions for all except the 5 nm target and, even there, the laser will snowplow and push the preplasma back into the target. This is a coarse indicator only, I think.

C) There should be very strong magnetic fields present in the target. Are there and was it okay to neglect them in the discussion?

D) Please specify the laser pulse energy! From the simulation intensity, I guess it was hundreds of mJ.

E) I don't understand the 'timing of transparency' plot in Fig. 6d. How was this curve derived? It seems to be saying that at near zero laser amplitude, transparency occurs early. Is this really just a plot of 'a' versus time?

F) I recommend you write 'log' for the logarithm instead of 'lg' in Fig. 4. The lambda didn't print correctly in my pdf in Fig. 6a.

Reviewer #2 (Remarks to the Author):

Review

The manuscript establishes a novel experimental scheme for accelerating electrons at relativistic energies with the help of high-intensity lasers. The acceleration mechanism is vacuum laser acceleration (VLA) which is experimentally challenging because the injected electrons into the laser field must have specific initial conditions that are difficult to control. The conditions for VLA are reached by preaccelerating electrons during the particular laser plasma interaction regime called relativistic transparency (RT) regime. This short duration regime requires extremely high contrast and ultraintense lasers as well as extremely thin targets (a few tens of nanometer thickness). In this way, the well-written manuscript presents a complete study combining experimental results with careful interpretations of 2D-PIC simulations that convincingly support the experiment.

In my opinion, although some points have to be addressed in the paper (see below), and if the quality of this work deserves a high impact review, it is not clear if it is appropriate for Nature Communications. Indeed, this work does not claim to present a breakthrough in the field of electron acceleration (even if

VLA is difficult to obtain and to prove). The reader has difficulty in realizing how this scheme is better or more effective than others already published. For example, in the introduction, it is stated that "clear experimental demonstration of VLA has been difficult" (with three references) without explaining what was controversial in these experiments, an explanation that would reinforce the added value of the authors' work. More importantly, the "breakthrough" in the VLA domain has been proved in the cited Ref. 25 (published in Nature Physics in 2016) and, comparatively, it seems to me that the method set out in Ref. 25 is more elegant and efficient than the one requiring the relativistic transparency effect. From the experimental point of view, (i) the plasma mirror technique for enhancing the contrast of the laser pulse (used in Ref. 25) is easier and better controlled than the RT regime (the shot to shot reproducibility in Ref. 25 is very good). (ii) It does not need any target. (iii) The maximum accelerated energy is about 10 to 15 MeV, less than 20 MeV demonstrated in this work, but at an intensity on target 10 times less than reported here. So, I understand that the purpose of the reported experiment is not necessary to compare with past experiences; because, on the one hand, of the lack of "breakthrough", but, on the other hand, of the quality of this work, I would recommend the manuscript for publication (after some corrections, see below) in a high impact journal but not necessarily Nat. Communications. Note that if the Editor considers it admissible for Nat. Comm., I will not oppose it.

Following are three points that need to be clarified and/or improved.

(1) Experimental setup: as noticed by the authors, it is very difficult to ensure that the VLA regime is experimentally achieved. The authors present convincing measurements attesting that the contrast of the laser is high enough to reach the transparency regime, but the precise conditions of interaction remain unclear. For example, the authors claim to use a laser delivering an energy up to 10 J in 50 fs in a 2 μm diameter focal spot, thus providing an intensity on target well above the one chosen for the PIC simulations ($2.1 \times 10^{20} \text{ W/cm}^2$). Even if the focal spot of 2 μm contains 1J, that is to say 10% of the total laser energy, the intensity should reach $6.3 \times 10^{20} \text{ W/cm}^2$ which is still 3x higher than what is chosen in the simulations. It is known that the characterization of micrometer size focal spots is difficult especially with regard to a PW facility. But the intensity on target is a crucial parameter for this kind of experiment. As a consequence, I believe that the measurement of the intensity and/or of the energy encircled in the 2 μm diameter focal spot should be presented and clarified, for example (partly) in the Supplementary Information part.

(2) The transition to the VLA regime with respect to the foil thickness is not clear in the manuscript. Figure 2 shows that 20 nm thick targets are opaque to the laser. This is in contradiction with what is written in the chapter "Modeling of the experimental results" ["The two-dimensional (2D) simulations adopting experiment-like conditions show qualitatively the same physics of electron acceleration for 5~30 nm foils...]. Also, PIC simulations demonstrating VLA (requiring relativistic transparency) are done with targets having 20 nm thickness. The lack of precision in the estimation of this thickness is annoying and is indeed commented in chapter 3 of Supplementary Information. The authors invoke vapor contamination increasing the actual thickness; wouldn't another missing link be the actual value of the laser intensity? Also, would it be possible to measure the actual thickness before the laser shot (in

vacuum) by using an interferometric method? As this point is important for the experiment, I think that at least a part of the discussion regarding the actual thickness should be included into the main text (and not only in the Supplementary Information).

(3) The chapter "Stochastic electron acceleration in front of the foil and RT injection" is not clear to me. I am not expert in numerical simulations but I guess this must be the case for many readers of Nat. Communications... Is it possible to better explain this part by emphasizing on what you want to demonstrate and to see in the simulations (and in Figure 6)? On this subject, it would be suitable for the reader to physically define what represents the Poincaré-map analysis (regarding the importance of the injection phase matching of the electrons into the laser field).

Finally, a minor correction: in chapter "Relativistic transparency injection", there is probably a missing word (heating, acceleration?) in the sentence "...where the refluxing electrons undergo a violent stochastic ?? (discussed in detail later)...".

Reviewer #3 (Remarks to the Author):

The authors present experimental evidence of a new injection method for electron acceleration by a short pulse, high-intensity laser in vacuum. In contrast to a recently demonstrated scheme based on a plasma mirror to inject and accelerate electrons with a reflected laser pulse, the method presented here relies on an optical switch based on relativistic transparency to capture and boost the energies of electrons transmitted through a thin foil.

Overall this appears to be a well executed experiment with multiple complementary diagnostics and which has been carefully analyzed with the help of PIC simulation and other particle tracking tools. On the other hand, one could argue that there is a heavy reliance on the simulation tools to support the authors' main contention that relativistically induced transparency is key to explaining the 2x increase in electron energies observed in the spectra for the 5nm foils compared to all other thicknesses (20-2000nm). Besides the spectra, the other diagnostics offered are a change in transmitted electron transverse beam profile, and a correlation of maximum electron energy with laser transitivity.

The authors' simulations show that significant additional vacuum acceleration of electrons transmitted through the foil occurs when its density falls below the relativistically corrected critical density. This 3rd phase of the interaction is comparable to the plasma mirror scheme of Ref. 25, so the question naturally arises of what advantages this transmissive scheme would have over the reflective one?

Similar super-ponderomotive VLA has also been observed with nanocluster targets - see Cardenas et al., Sci. Rep. 9, 7321 (2019). Phase-locked injection also plays a role in this scenario, which is also aided by field amplification at the target surface.

In summary I think this is a solid piece of work which I cannot find serious fault with. It is well written and argued and can be published provided the authors consider the points mentioned above.

Response to reviewer comments

Here, we provide a point-by-point response (in blue) to each reviewer (with their comments in black). The corresponding revision in the manuscript is briefly summarized in each response.

Texts quoted from the manuscript are presented in *italic* style; Please refer to the revised main manuscript and Supplementary Information for the complete revisions where changes are highlighted in the red color.

Reviewer #1 (Remarks to the Author):

Review

The manuscript presents an important finding on relativistic transparency (RT). Namely, dramatic electron spectra and angular distributions are presented that suggest the onset of RT, which is still a relatively unexplored phenomenon **experimentally**. The experimental results combined with PIC simulations also indicate a novel electron acceleration mechanism at play.

The experiment seems well performed and the results are striking. I have some concerns about the analysis, although I expect they can be addressed. This work is described in sufficient detail that both experiment and simulation could be reproduced. The subject area and findings are appropriate for Nat. Comm. The writing was conversational and generally straightforward to follow.

Discussion

Relativistic transparency is a key phenomenon in a wide range of intense laser-plasma experiments. It is of fundamental importance because it is an example of the interaction of light with a relativistic medium and not just isolated relativistic particles. I think this makes this area of broad interest. Surprisingly, however, there are relatively few experiments that directly measure this effect, with reliance on PIC simulations usually the primary means of identifying its presence. Although PIC plays a key role here, the data are compelling, as summarized in Fig. 2. The difference between the 20 nm and 5 nm thick film spectra is suggestive. The concomitant change in the angular distribution is also persuasive. I don't see any flaws in the performance of the experiment. It may seem surprising that such an experiment has not been performed already with ultrashort lasers (as opposed to lasers like Trident or Phelix), but few such lasers have the contrast needed (at least in the US) and the targets are harder to work with.

Re: We thank the reviewer for taking the time to evaluate our manuscript, and for the positive assessment of the importance, novelty, and quality of our work.

I disagree somewhat with the motivation provided for this work. The primary justification given seems to be development of a source of accelerated electrons. However, laser plasma

accelerators (eg. wakefield) provide a far better electron beam in energy, spectrum, yield and angular distribution and with far less restrictive requirements on both the laser and target. However, I find their argument that these results provide new insight on RT persuasive and sufficient. Stochastic electron dynamics have long been identified and the importance of the target in dephasing accelerated electrons is well known (ie. JxB). However, the mechanism here depends on shaping of the electron momentum distribution in front of the target due to the transient standing wave followed by achievement of RT and transmission of the laser through the target with injected electrons. Substantial acceleration occurs far from the target.

Re: We completely agree with the reviewer that the wakefield acceleration produces better-quality beams and our work provides new insight on the RT plasmas. Though our work provides new insight in to the RT plasmas, VLA has the potential to generate micro-Coulomb electron charge per pulse, which is necessary for developing laser-driven MeV x-ray sources for high-areal-density-object radiography [see, for example, S. Palaniyappan et al. *Laser and Particle Beams*, 36, 502-506, (2018)]. Nevertheless, following the reviewer's comment, we have added the following statement in the 'Introduction' section of the manuscript.

“More importantly, as we shall see, our results provide new insights into the electron dynamics in RT plasmas. Due to its volumetric interaction nature, the RT regime has been widely recognized as a compelling platform for laser ion accelerators, x-ray sources, and relativistic optics [8, 9, 11, 12]. However, most of the previous studies have focused on these secondary sources leaving their primary driver—fast electrons—least understood. By advancing the understanding in the laser-electron coupling, our work should stimulate new developments in the various secondary sources.”

Concerns

The PIC simulations play a critical role in this work and, indeed, their discussion takes up most of the manuscript. I do think the primary conclusions derived from the simulations are likely correct, but I have some concerns.

1) The biggest concern is the use of 2D simulations. Although I am not personally familiar with SMILEI, the PIC code used, 3D simulations of an experiment with these parameters is not out of reach any more. (I do recognize a bigger grid is needed to see the acceleration.) Still, use of 2D here is surprising. As the authors are aware, there are several issues. 2D tends to exaggerate the density of the target as it heats and, of course, the density is critical to RT. Also, 2D can greatly exaggerate and extend the sheath field which will affect the injection of the electrons into the pulse. Finally, although I think this a small effect here, the phase and amplitude of a 2D Gaussian approaches the target differently than that for 3D. However, it is stated that a full 3D simulation was indeed performed with very similar results. It is surprising these weren't used for the analysis, then. In any case, they should be shown, preferably in a figure corresponding to Fig. 2.

Re: We thank the reviewer for the thoughtful comment. As mentioned by the reviewer, 3D PIC simulations can have differences from 2D PIC simulations and the 3D simulations of this kind could be demanding. In fact, in our case, the 3D simulations would require a very high resolution to resolve the very thin target on the one hand, and a big grid for VLA to develop

sufficiently in the rear space on the other hand. The resulting data storage (including particle tracking) and computation are huge and not easy to manage.

Nevertheless, we have followed the reviewer's advice and provided a dedicated section (the last section "Validation using 3D PIC simulations" before "Outlook") on the 3D simulations including a new Fig. 8 (which corresponds to Fig. 2). We have also provided additional information on the 3D particle tracking (Fig. 8a) and target dynamics (Fig. 8d). As one can see, these 3D simulations demonstrated very similar results (in terms of the energy spectrum, beam profile, acceleration dynamics, and the onset of RT) compared with their 2D counterparts. Thus, the 3D simulations validated our interpretation from the 2D simulations.

As mentioned by the reviewer, we do see differences between 2D and 3D simulations in terms of the density evolution and laser dynamics. These differences are discussed in detail in the above-mentioned section. Additional details are also presented in the Supplementary Information. It is worth noting that these differences are quantitative, and they do not alter the RT-based VLA physics that we have put forward in this work.

2) The experiment shows a dramatic difference between the 20 nm and 5 nm targets. I think that is a primary strength of the paper, in fact. However, the discussion suggests the simulations don't see a dramatic difference (see also Fig S3) and, indeed, 20 nm is used to compare to the experimental 5 nm. This should be discussed. I'm guessing the electron spectra differed significantly between 5 nm and 20 nm in the simulations. Was the thickness used as a tuning parameter? If so, this doesn't necessarily invalidate the major claims, but we should be informed.

Re: Thanks for the comment. Yes, we do not see a similar contrast between 5 nm and 20 nm in the present simulations as found in experiments. For the laser (peak amplitude, duration) and plasma parameters used, both the 5 nm and 20 nm cases belong to the RT regime in the simulation, whereas in experiments the 20 nm falls in the opaque regime where no laser transmission is observed. There are challenges in quantitatively reproducing all aspects of the experimental data in the simulation by following exactly the nominal target thickness. Several uncertainties including the target fabrication (as informed by the manufacturer), vapor contamination, and on-target laser intensity and focal-spot quality have contributed to this difficulty. For example, despite the estimated intensity in the experiment, all the simulations that we did point towards a lower a_0 when comparing the experimental results over a wide range of parameters such as backscattered spectra, electron spectra, and the amount of laser energy transmitted through the target. Since the intensity is only an estimated quantity and is not directly measured but often assessed with simulations, we believe it is appropriate to stick with the lower a_0 . This is also further complicated by the quantitative difference between 2D and 3D simulations. We use 2D simulations to infer the electron dynamics since deciphering the data from a 3D simulation is a daunting task. Despite the differences, we believe the physics we described in the manuscript applies to a wide range of laser and plasma parameters within the regime of relativistic transparency irrespective of the specific laser-target parameters (please see the supplemented Fig. S4 and S5.) We will continue to explore this difference in the future.

In response to the reviewer's comment, we have detailed the above considerations in the Supplementary Information in Fig. S4. We have also revised the main text in the first paragraph

of the section “Modeling of the experimental results” to briefly summarize our key points, as quoted below:

“The simulations are mainly used to elucidate the key physics that underpin the significantly enhanced acceleration observed in the RT regime. Quantitatively reproducing all observables using the nominal laser-target conditions is subject to the uncertainties in the target fabrication, vapor contamination, and on-target laser intensity, and focal spot quality. Therefore, we have explored a range of foil thickness and laser intensity closely around the nominal parameters, and these simulations show qualitatively the same physics of electron acceleration (detailed below) for 5~30 nm foils once the RT sets in, despite varying amounts of laser transmission and electron acceleration (see supplemented Fig. S4 and Fig. S5). Without loss of generality, we take the 20 nm and 200 nm case as representative of the RT and opaque regime, respectively, and focus on identifying the acceleration mechanism pertinent to the RT regime.”

The remainder of my concerns are lesser and, I think, readily addressed.

A) I disagree with the simple model explaining spectral broadening (page 4, merged pdf). It makes it seem as if it is all simple Doppler shifts, whereas phase modulation due to ionization and, especially, density variation, is just as strong.

Re: Yes, we agree with the reviewer that besides the Doppler shift, spectral broadening could also be introduced via ionization-driven phase modulation. The instantaneous frequency shift is given as $\Delta\omega = (\omega_{inst.} - \omega_0) = -\delta\phi/\delta t = -\frac{\omega_0}{c} \int \frac{\delta\eta}{\delta t} dx$, where the relativistic corrected plasma refractive index ($\eta = \sqrt{(1 - n_e/\gamma n_c)}$), n_e , is the electron density, n_c is the plasma critical density, $\gamma = \sqrt{1 + a_0^2/2}$ is average Lorentz factor of electron and a_0 is laser strength parameter. An increase in the electron density (by ionization) will induce a blue shift ($\frac{\delta n_e}{\delta t} > 0$; $\frac{\delta\eta}{\delta t} < 0$; $\Delta\omega > 0$) in the spectrum, whereas a decrease in the electron density (via recombination) will induce ($\frac{\delta n_e}{\delta t} < 0$; $\frac{\delta\eta}{\delta t} > 0$; $\Delta\omega < 0$) redshift. At this intensity and short time-scale, we can safely rule out any recombination-driven red-shift. Regarding the ionization driven blue shift, all the ionization state of the target species (C and H) saturates at the order of magnitude lower intensity level of our peak laser intensity ($> 10^{20} Wcm^{-2}$). For instance, C^{6+} can be obtained via field ionization at the intensity of $6.4 \times 10^{18} Wcm^{-2}$, which is nearly two orders of magnitude smaller than our peak laser intensity. This implies that much before the peak of our laser pulse arrival, the ionization state of all the species in the focal spot was already saturated and therefore the ionization-related phase modulation does not affect much the spectral broadening of the main pulse.

Besides ionization-driven phase shift, we may have a strong influence of relativistic mass effect. At the rising part of the laser pulse, the γ increase which causes an increase of the plasma refractive index ($\eta = \sqrt{(1 - n_e/\gamma n_c)}$), and therefore introduces a redshift ($\frac{\delta\gamma}{\delta t} > 0$; $\frac{\delta\eta}{\delta t} > 0$; $\Delta\omega < 0$). After passing the peak of the laser pulse, the γ decrease which causes a decrease of the plasma refractive index and therefore introduces a blue shift ($\frac{\delta\gamma}{\delta t} < 0$; $\frac{\delta\eta}{\delta t} < 0$; $\Delta\omega > 0$). Since our laser pulse is symmetric in time, we are observing quite symmetric blue and red spectral broadening.

In response to the reviewer's comment, we have now:

1) Modified the following line in the main text as:

“This result, as a coarse indicator, could imply negligible pre-pulse effects and the interaction of the main pulse with a high-density plasma even for the thinnest 5 nm target.”

2) Added the following lines in the Supplementary Information when discussing Fig. S2:

“In our experimental conditions, the spectral broadening could be introduced via relativistic mass effect. The instantaneous frequency shift is given as $\Delta\omega = (\omega_{inst.} - \omega_0) = -\delta\phi/\delta t = -\frac{\omega_0}{c} \int \frac{\delta\eta}{\delta t} dx$, where the relativistic corrected plasma refractive index ($\eta = \sqrt{(1 - n_e/\gamma n_c)}$), n_e , is the electron density, n_c is the plasma critical density, $\gamma = \sqrt{1 + a_0^2/2}$ is average Lorentz factor of electron and a_0 is laser strength parameter. At the rising part of the laser pulse, the γ increase which causes an increase of the plasma refractive index ($\eta = \sqrt{(1 - n_e/\gamma n_c)}$), and therefore introduces a redshift ($\frac{\delta\gamma}{\delta t} > 0$; $\frac{\delta\eta}{\delta t} > 0$; $\Delta\omega < 0$). After passing the peak of the laser pulse, the γ decrease which causes a decrease of the plasma refractive index and therefore introduces a blue shift ($\frac{\delta\gamma}{\delta t} < 0$; $\frac{\delta\eta}{\delta t} < 0$; $\Delta\omega > 0$). Since our laser pulse is symmetric in time, we are observing quite symmetric blue and red spectral broadening.”

B) It is emphasized that the independence of the spectra on target thickness is clear evidence of good contrast. Why is this true? The front surface physics will be the same under these conditions for all except the 5 nm target and, even there, the laser will snowplow and push the preplasma back into the target. This is a coarse indicator only, I think.

Re: We thank the reviewer for the comments on this point and agree that our back-reflection spectrum diagnostic, due to time-integration, may provide only a coarse indicator. Here, by ‘high-contrast’, we mean our laser contrast is good enough to prevent any prepulse-induced target deterioration, especially for the ultrathin, few nanometer foils. In presence of strong prepulse, the thin targets could be more susceptible to prepulse-driven foil expansion, as there is only finite mass available.

Moreover, it is also possible that the laser can snowplow and push the preplasma back into the target, which will cause a larger redshift in comparison with opaque foils. However, we observe that the nature of the back-reflected spectrum is similar for all the targets, regardless of thickness. Based on this, we could infer that the target front surface interaction condition remained similar down to 5nm thin foil.

In response to this comment, we have now added the following lines in the Supplementary Information (end of the first paragraph on page 2):

“The observation of back-reflected spectrum being similar for all the targets regardless of thickness as a coarse indicator could imply that the laser contrast is good enough to prevent any prepulse induced target deterioration, especially for the ultrathin, few nanometer foils. In

presence of strong prepulse, the thin targets could be more susceptible to prepulse driven foil expansion, as there is only finite mass available.”

C) There should be very strong magnetic fields present in the target. Are there and was it okay to neglect them in the discussion?

Re: Yes, magnetic fields, especially the transverse components, can be excited in the target once electrons get accelerated. However, these fields are mainly contained in the bulk of the target and their effects on the acceleration should be limited to the short electron transit time (i.e., the time needed for electrons to escape the plasma target), mostly affecting the front surface acceleration and the ensuing sheath deceleration. Their exact contributions would be complicated due to the co-existence of laser fields and self-generated electric/magnetic fields. In fact, as we mentioned in the paper, the stochastic acceleration (purely due to the standing-wave laser fields) from the test-particle simulation would account for 75% of the front-side acceleration as observed in PIC simulations. The rest should be assisted by these self-generated fields. We have revised this sentence (adding the contribution of magnetic fields) close to the end of the last paragraph in the section “Stochastic electron acceleration in front of the foil and RT injection” which is quoted below:

“This result accounts for about 75% of the front-side acceleration as observed in the PIC simulation, with the rest of the acceleration possibly assisted by the self-generated electric and magnetic fields near the bulk of the target”

Fig. R1. Information (top: longitudinal momentum p_x ; middle/bottom: electric/magnetic fields experienced) of a typical VLA electron obtained from 3D particle tracking for the RT regime. The red vertical line in the top panel shows the original position of the target.

The effects of these magnetic fields on VLA once laser transmits through should be more limited because the latter takes in the rear space relatively far away from the bulk target. This

is illustrated in Fig. R1; the rear-space acceleration takes beyond $x = 15\lambda_0$, where the fields are dominated by the E_y, B_z components which are the P-polarized laser fields.

D) Please specify the laser pulse energy! From the simulation intensity, I guess it was hundreds of mJ.

Re: Thanks for pointing this out. The laser energy on target was 6.8 J. The focal spot with Gaussian FWHM of 1.7 μm , containing 32% of the total energy. Now we have specified these parameters in the section “Experimental setup” of Methods and included detailed characterization of the laser focal spot at the beginning of the Supplementary Information (Fig. S1).

E) I don't understand the 'timing of transparency' plot in Fig. 6d. How was this curve derived? It seems to be saying that at near zero laser amplitude, transparency occurs early. Is this really just a plot of 'a' versus time?

Re: Thanks for raising this issue. It is very useful for us to further clarify this plot. Yes, it is just a plot of laser envelope 'a' versus time. It should be interpreted this way: For a given transparency timing, one would have a corresponding laser amplitude for the stochastic acceleration following the dashed curve. Then through the solid curve, the laser amplitude is mapped to the maximum p_x that can be extracted from the stochastic acceleration. In other words, if the transparency occurs too early or too late, the laser amplitudes available for the stochastic acceleration would be small and the transparency injection would be less efficient.

To clear any ambiguities, we have revised relevant discussions (the end of the paragraph before the section 'Vacuum laser acceleration in the transmitted laser pulse') which are quoted below:

“Moreover, the short timescale involved suggests that the effective laser amplitude for stochastic acceleration can be mapped to the timing of the onset of RT (Fig. 6d). For example, if the transparency occurred too early/late (right vertical axis), the corresponding laser amplitude (horizontal axis) and the maximum p_x from the stochastic acceleration (left vertical axis) would be small.”

F) I recommend you write 'log' for the logarithm instead of 'lg' in Fig. 4. The lambda didn't print correctly in my pdf in Fig. 6a.

Re: Thanks for the suggestion, and we have fixed them in the revised figures.

Reviewer #2 (Remarks to the Author):

Review

The manuscript establishes a novel experimental scheme for accelerating electrons at relativistic energies with the help of high-intensity lasers. The acceleration mechanism is vacuum laser acceleration (VLA) which is experimentally challenging because the injected electrons into the laser field must have specific initial conditions that are difficult to control. The conditions for VLA are reached by preaccelerating electrons during the particular laser plasma interaction regime called relativistic transparency (RT) regime. This short duration regime requires extremely high contrast and ultraintense lasers as well as extremely thin targets (a few tens of nanometer thickness). In this way, the well-written manuscript presents a complete study combining experimental results with careful interpretations of 2D-PIC simulations that convincingly support the experiment.

Re: Thanks very much to the reviewer for taking the time to review our work, and for the positive views on the novelty and quality of our work.

In my opinion, although some points have to be addressed in the paper (see below), and if the quality of this work deserves a high impact review, it is not clear if it is appropriate for Nature Communications. Indeed, this work does not claim to present a breakthrough in the field of electron acceleration (even if VLA is difficult to obtain and to prove). The reader has difficulty in realizing how this scheme is better or more effective than others already published. For example, in the introduction, it is stated that "clear experimental demonstration of VLA has been difficult" (with three references) without explaining what was controversial in these experiments, an explanation that would reinforce the added value of the authors' work. More importantly, the "breakthrough" in the VLA domain has been proved in the cited Ref. 25 (published in Nature Physics in 2016) and, comparatively, it seems to me that the method set out in Ref. 25 is more elegant and efficient than the one requiring the relativistic transparency effect. From the experimental point of view, (i) the plasma mirror technique for enhancing the contrast of the laser pulse (used in Ref. 25) is easier and better controlled than the RT regime (the shot to shot reproducibility in Ref. 25 is very good). (ii) It does not need any target. (iii) The maximum accelerated energy is about 10 to 15 MeV, less than 20 MeV demonstrated in this work, but at an intensity on target 10 times less than reported here. So, I understand that the purpose of the reported experiment is not necessary to compare with past experiences; because, on the one hand, of the lack of "breakthrough", but, on the other hand, of the quality of this work, I would recommend the manuscript for publication (after some corrections, see below) in a high impact journal but not necessarily Nat. Communications. Note that if the Editor considers it admissible for Nat. Comm., I will not oppose it.

Re: We thank the reviewer for the comments that helped us further clarify the significance of our work. We agree with the reviewer that the plasma mirror technique is clearly a more elegant way of demonstrating the VLA process. Our objective here is to understand the generation of super-ponderomotive electrons from the relativistic transparent plasmas and not compete with the plasma mirror scheme. Nevertheless, compared to the plasma mirror scheme, the transmission scheme in the relativistic transparent plasmas discussed here is a qualitatively a new scheme. And it is also a more suitable scheme for the secondary

bremstrahlung MeV x-ray generation and proton/neutron beam generation. Since relativistic transparency is a volumetric process as opposed to the plasma mirror scheme where the electrons are essentially peeled off from the surface, we expect the laser-to-electron conversion efficiency scale faster when using the relativistic transparent plasmas driven by lasers with large focal volume. In the past, we have observed 5-10 μC of electrons from sub-micron thick foils using the Trident laser at the Los Alamos National Laboratory [J. A. Cobble et al. Physics of Plasmas 23,093113, (2016)]. Several experiments have shown that sub-micron foils undergo relativistic transparency when driven by the Trident laser [S. Palaniyappan et al. Nature Physics, 8, 763-769 (2012), D. Jung et al. Physics of Plasmas 20,083103 (2013)]. To the authors' best knowledge, the 5-10 μC charge is probably the highest electron charge measured from a laser-driven accelerator. The high charge is essential for generating a high MeV x-ray dose for radiography applications [see, for example, S. Palaniyappan et al. Laser and Particle Beams, 36, 502-506, (2018)]. With the new understanding of how the super-ponderomotive electrons originate from relativistic transparent plasmas, we believe we could increase the laser-to-electron conversion efficiency further by manipulating the laser polarization and plasma target that would enhance the secondary sources.

In response to the reviewer's comment, We have revised the text close to the end of the "Introduction" to highlight the above points and to emphasize the novelty and significance of our work, quoted here as:

"Compared with the plasma mirror scheme based on a surface interaction, the present transmission scheme involves the volumetric RT process which may imply faster scaling of laser-to-electron conversion efficiency when driven by lasers with larger focal volume. More importantly, as we shall see, our results provide new insights into the electron dynamics in RT plasmas. Due to its volumetric interaction nature, the RT regime has been widely recognized as a compelling platform for laser ion accelerators, x-ray sources, and relativistic optics [8, 9, 11, 12]. However, most of the previous studies have focused on these secondary sources while leaving their primary driver—fast electrons—least understood. By advancing the understanding in the laser-electron coupling, our work should stimulate new developments in the various secondary sources."

Finally, following the reviewer's advice, we have added in the "Introduction" (middle of the second paragraph) more discussions on the previous VLA experiments, quoted below as:

"Despite using relativistically intense lasers, many experiments demonstrated only 100s keV acceleration [19]. While the scheme of ionizing highly-charged ions predicted GeV acceleration numerically [34], preliminary experiments only showed ~ 1 MeV photoelectrons generated from this process [36]. The scheme exploiting a partially transmitted laser for acceleration between two foils showed no amplification in the energy but only an increase of electron number around 1 MeV energy [37]."

With the above clarifications and revisions, we hope that the literature is clear, the value of our work is distinguished, and the manuscript now meets the standards of Nature Communications.

Following are three points that need to be clarified and/or improved.

(1) Experimental setup: as noticed by the authors, it is very difficult to ensure that the VLA regime is experimentally achieved. The authors present convincing measurements attesting that the contrast of the laser is high enough to reach the transparency regime, but the precise conditions of interaction remain unclear. For example, the authors claim to use a laser delivering an energy up to 10 J in 50 fs in a 2 μm diameter focal spot, thus providing an intensity on target well above the one chosen for the PIC simulations ($2.1 \times 10^{20} \text{ W/cm}^2$). Even if the focal spot of 2 μm contains 1J, that is to say, 10% of the total laser energy, the intensity should reach $6.3 \times 10^{20} \text{ W/cm}^2$ which is still 3x higher than what is chosen in the simulations. It is known that the characterization of micrometer size focal spots is difficult especially with regard to a PW facility. But the intensity on target is a crucial parameter for this kind of experiment. As a consequence, I believe that the measurement of the intensity and/or of the energy encircled in the 2 μm diameter focal spot should be presented and clarified, for example (partly) in the Supplementary Information part.

Re: Thanks for pointing this out. Considering the ALEPH laser at the Colorado State University is relatively new to come online, the laser parameters are not characterized as accurately as we would have like it to be at the time of our experiment. Since then, the laser team has gone to great lengths to characterize the system. As per reviewer's suggestion, we have added a description of the laser intensity characterization in the Supplementary Information (Fig. S1). It is briefly summarized here: during the experiment, the frequency-doubled (400 nm) laser, of 75 fs pulse duration and maximum of 6.8 Joule, was focused with a spot size of 1.7 μm (FWHM), having 32% concentration of the total laser energy. The resultant estimated peak intensity reaches $0.9 \times 10^{21} \text{ Wcm}^{-2}$. (The pulse duration was earlier claimed to be 50fs based on auto-correlation measurements. However, very recent measurements of the pulse shape using a Self-Diffraction-Frequency-Resolved-Optical-Gating (SD-FROG) show that the pulse is 75fs long. We do not believe this changes the qualitative explanation provided in our manuscript.)

This nominal intensity is indeed higher than what we used for PIC simulations. Partly in response to Reviewer 1's comment, we found that all the simulations that we performed pointed towards a lower a_0 when compared to the experimental results over a wide range of parameters such as backscattered spectra, electron spectra, and the amount of laser energy transmitted through the target. Since the intensity is only an estimated quantity and is not directly measured but often assessed with simulations, we believe it is appropriate to stick with the lower a_0 . In addition, considering the differences between 2D and 3D simulations, we believe that the laser intensity requires further laser characterization in the future. Despite the differences, we believe the physics we described in the manuscript applies to a wide range of laser and plasma parameters within the regime of relativistic transparency irrespective of the specific laser-target parameters. To see this, we have provided supplemented Fig. S4 and S5; in particular, Fig. S5 shows the results for the nominal intensity (or amplitude $a_0=10$) where qualitatively the same acceleration mechanism is identified for the RT regime. We will continue to explore this difference in peak intensity in the future.

(2) The transition to the VLA regime with respect to the foil thickness is not clear in the manuscript. Figure 2 shows that 20 nm thick targets are opaque to the laser. This is in contradiction with what is written in the chapter "Modeling of the experimental results" ["The two-dimensional (2D) simulations adopting experiment-like conditions show qualitatively the same physics of electron acceleration for 5~30 nm foils..."]. Also, PIC simulations

demonstrating VLA (requiring relativistic transparency) are done with targets having 20 nm thickness. The lack of precision in the estimation of this thickness is annoying and is indeed commented in chapter 3 of Supplementary Information. The authors invoke vapor contamination increasing the actual thickness; wouldn't another missing link be the actual value of the laser intensity? Also, would it be possible to measure the actual thickness before the laser shot (in vacuum) by using an interferometric method? As this point is important for the experiment, I think that at least a part of the discussion regarding the actual thickness should be included into the main text (and not only in the Supplementary Information).

Re: Thanks a lot for raising this comment. Yes, the uncertainty in actual on-target laser intensity is another major factor that makes quantitative comparisons using the nominal target thickness difficult. This is complicated further by the quantitative differences between the 2D and 3D simulations.

We also thank the reviewer for the suggestion of pre-characterizing target thickness with interferometry. However, during our experiment, due to space constraints and configuration of experimental set up in and around the chamber, it was not possible to add this diagnostic.

Following the reviewer suggestion, we have explicitly included in the main text the above key considerations that have led us to the choice of this set of target thickness for simulations; please see the beginning of the paragraph following "Modeling of the experimental results", which is quoted below:

"The simulations are mainly used to elucidate the key physics that underpin the significantly enhanced acceleration observed in the RT regime. Quantitatively reproducing all observables using the nominal laser-target conditions is subject to the uncertainties in the target fabrication, vapor contamination, and on-target laser intensity, and focal spot quality. Therefore, we have explored a range of foil thickness and laser intensity closely around the nominal parameters, and these simulations show qualitatively the same physics of electron acceleration (detailed below) for 5~30 nm foils once the RT sets in, despite varying amounts of laser transmission and electron acceleration (see supplemented Fig. S4 and Fig. S5). Without loss of generality, we take the 20 nm and 200 nm case as representative of the RT and opaque regime, respectively, and focus on identifying the acceleration mechanism pertinent to the RT regime."

(3) The chapter "Stochastic electron acceleration in front of the foil and RT injection" is not clear to me. I am not expert in numerical simulations but I guess this must be the case for many readers of Nat. Communications... Is it possible to better explain this part by emphasizing on what you want to demonstrate and to see in the simulations (and in Figure 6)? On this subject, it would be suitable for the reader to physically define what represents the Poincaré-map analysis (regarding the importance of the injection phase matching of the electrons into the laser field).

Re: Thanks for the comments that help us to clarify our paper. We have followed the reviewer's advice and revised the text at the beginning of the two paragraphs in this chapter. The revision in response to "what we want to demonstrate and see in the simulations" is quoted below as:

"The ensuing sharp reversal of p_x in front of the foil (Fig. 5b) involves electron interaction with a complicated field configuration including both the laser fields and self-generated plasma

fields. To see what causes the front-side acceleration, we perform simplified test-particle simulations (see the Methods for detailed setup) where the above complicated fields can be separated easily and added back one by one.”

We have also added a sentence to define what represents the Poincare-map analysis:

“To further clarify the acceleration, we transform these dynamics into a Poincare map (see the Methods for details) which represents the traces of the electrons on a phase-space cross-section that is commonly used to distinguish the stability (i.e., stochastic or regular motion) of their quasi-periodic orbits.”

The “phase-matching” for the stochastic acceleration is also commented in the revised sentence (second paragraph of this chapter) quoted below:

“That is, these electrons may be briefly in phase with either of the lasers due to perturbation by the other laser field and be randomly accelerated to any point of the area in a few laser cycles.”

Finally, a minor correction: in chapter "Relativistic transparency injection", there is probably a missing word (heating, acceleration?) in the sentence "...where the refluxing electrons undergo a violent stochastic?? (discussed in detail later) ...".

Re: Thanks for pointing this out. We have now added the missing word.

Reviewer #3 (Remarks to the Author):

The authors present experimental evidence of a new injection method for electron acceleration by a short pulse, high-intensity laser in vacuum. In contrast to a recently demonstrated scheme based on a plasma mirror to inject and accelerate electrons with a reflected laser pulse, the method presented here relies on an optical switch based on relativistic transparency to capture and boost the energies of electrons transmitted through a thin foil.

Overall this appears to be a well executed experiment with multiple complementary diagnostics and which has been carefully analyzed with the help of PIC simulation and other particle tracking tools. On the other hand, one could argue that there is a heavy reliance on the simulation tools to support the authors' main contention that relativistically induced transparency is key to explaining the 2x increase in electron energies observed in the spectra for the 5nm foils compared to all other thicknesses (20-2000nm). Besides the spectra, the other diagnostics offered are a change in transmitted electron transverse beam profile, and a correlation of maximum electron energy with laser transitivity.

Re: Thanks very much to the reviewer for taking the time to evaluate our work, and for the positive views on the novelty and quality of our work.

The authors' simulations show that significant additional vacuum acceleration of electrons transmitted through the foil occurs when its density falls below the relativistically corrected critical density. This 3rd phase of the interaction is comparable to the plasma mirror scheme of Ref. 25, so the question naturally arises of what advantages this transmissive scheme would have over the reflective one?

Re: Thanks for the insightful comments. There are two major advantages of our transmission scheme over the plasma mirror scheme. First, our scheme deals with volumetric plasma interaction instead of being limited to the surface of the plasma as in the plasma mirror scheme. Currently, we have only demonstrated the scheme using a tightly focused laser. With broader focal spots, we expect faster scaling of laser-to-electron conversion efficiency may be possible with the deeper involvement of the plasma. Second, our scheme deals with an important regime of laser-foil plasma interaction, i.e., the RT regime, which has important applications in laser ion accelerators, bremsstrahlung x-ray sources, and relativistic optics. This implies our scheme has broader potential applications in addition to being an electron source. In fact, most of the previous studies on RT plasmas have focused on these secondary sources and left the electron driver least touched. Our work on the detailed fast electron dynamics would point out potential ways to optimize the interaction and correspondingly those secondary sources.

Partly in response to the other reviewers, we have highlighted these points in the revised paper; please see the text close to the end of "Introduction" which is quoted below as:

"Compared with the plasma mirror scheme based on a surface interaction, the present transmission scheme involves the volumetric RT process which may imply faster scaling of laser-to-electron conversion efficiency when driven by lasers with larger focal volume. More importantly, as we shall see, our results provide new insights into the electron dynamics in RT plasmas. Due to its volumetric interaction nature, the RT regime has been widely recognized as a compelling platform for laser ion accelerators, x-ray sources, and relativistic optics [8, 9, 11, 12]. However, most of the previous studies have focused on these secondary sources while leaving their primary driver—fast electrons—least understood. By advancing the understanding in the laser-electron coupling, our work should stimulate new developments in the various secondary sources."

Similar super-ponderomotive VLA has also been observed with nanocluster targets - see Cardenas et al., Sci. Rep. 9, 7321 (2019). Phase-locked injection also plays a role in this scenario, which is also aided by field amplification at the target surface.

Re: Thanks for bringing to our attention this interesting publication. We have cited it as another example of demonstrating VLA in the "Introduction" as Ref. [35].

In summary I think this is a solid piece of work which I cannot find serious fault with. It is well written and argued and can be published provided the authors consider the points mentioned above.

Re: We would like to thank the reviewer again for the feedback.

REVIEWERS' COMMENTS

Reviewer #1 (Remarks to the Author):

Review

The authors have made a good faith and substantive effort to address my concerns regarding their first submission and I now believe their work is valid, significant, appropriate for Nat. Comm., and ready for publication.

Discussion

As I described in my previous review, the experimental work appears to be excellent. There were some flaws in the discussion, but those have been fixed. However, in this case, the results on their own do not constitute a high impact result and the ensuing analysis, including PIC simulations, is critical for assessing suitability for this journal. Broadly speaking, I felt that the authors had either overstated some aspects of their case or did not address complicating factors. The difference between 2D and 3D PIC and the target thickness dependence were two such issues. It was not so much that I thought there were flaws in the analysis as that I couldn't tell if there were. For example, 3D effects can be substantial for these interactions and the lack of 3D results when they said they had them was puzzling, but the new discussion is reassuring. I feel the authors have now addressed my concerns in their response to me and that they have modified the manuscript suitably. I also raised a number of technical concerns (interpretation of the back-reflected spectra, quasi-static B fields, etc) that were also addressed well. I think the paper is now significantly more nuanced and does a better job of clearly describing the uncertainties in their findings.

I read the other reviewers' reports with interest and I am in agreement with them. In particular, Reviewer #2 made excellent points and raised some of the same concerns that I did. One difference is that Reviewer #2 was perhaps less persuaded of the appropriateness of the work for this journal. This may be explained by somewhat different emphases in our reviews. As I explained in my first review, while I also had concerns with VLA as motivation, I focused on the relativistic transparency physics, finding that interesting and sufficient. I believe the revised introduction now does a much better job of motivating this work and placing it in perspective to other work on VLA and on possible applications. This is a significant improvement and removes my last concern.

In summary, I recommend this work for publication.

Reviewer #2 (Remarks to the Author):

Review

I find the authors' responses to the reviewers to be clear, precise and convincing. I understand that the reproduction by numerical simulations of an experiment as complex and characterized by parameters as difficult to control requires adjustments in the input data of the numerical code. It is therefore complicated to unravel the respective influences of the local laser intensity, the accurate thickness of the target or the 3D vs. 2D effects in PIC simulations. However, I believe this does not call into question the main purpose of the paper nor the seriousness of the results obtained. As a consequence, with some minor recommendations (see below), I think this manuscript will be suitable for publication in Nature Communications.

Minor corrections:

(1) The argument regarding the volumetric RT process involved in these experiments is convincing to stand out from other acceleration methods such as the wakefield acceleration. Thus, if the authors believe that VLA has the potential to accelerate micro-Coulomb electron charge bunches, i.e. a charge being about three orders of magnitude higher of what is frequently obtained in wakefield experiments, then it would be useful to state this in the introduction.

(2) The precision brought on the characteristics of the laser plasma interaction is welcome. I guess that the focal spot presented in Fig. S1 has been obtained for a low energy shot (some tens of micro-Joule) and not at full energy? If it is the case, this may be indicated in the figure caption (the intensity profile using full-laser amplification may differ from the one using only the pilot).

(3) I have small concerns regarding numerical values: (i) in Methods, the solid angle of the electron spectrometer is stated to $50 \mu\text{sr}$ that does not match the result of $\Delta\Omega = \Delta S/R^2 = 4 \text{ mm}^2/(500 \text{ mm})^2 = 16 \mu\text{sr}$; (ii) in S1, the (mean) laser intensity obtained with 32% of 6.8 J contained in a diameter of $1.7 \mu\text{m}$ and during 75 fs is $1.25 \cdot 10^{21} \text{ W/cm}^2$ (a quantity that is even further away from the one chosen in the simulations), and not the peak value of $0.9 \cdot 10^{21} \text{ W/cm}^2$ written in the paper; how is this peak intensity calculated?

(4) I would point out to the authors that the results presented in Ref. 19 (G. Malka et al.) have been the subject of controversies: see the two comments and replies of Kirk T. McDonald, Phys. Rev. Lett. 80, 1350 (1998) and of P. Mora and B. Quesnel, Phys. Rev. Lett. 80, 1351 (1998).

Reviewer #3 (Remarks to the Author):

The authors have provided a robust response to both my critique and that of the other referees, now including additional 3D simulations which seem to further corroborate the results. I see no reason to change my previous view that the work should be published.

Response to the reviewer's comments

Black: Reviewer comments

Blue: Our response

Reviewer Comments

Reviewer #1 (Remarks to the Author):

Review

The authors have made a good faith and substantive effort to address my concerns regarding their first submission and I now believe their work is valid, significant, appropriate for Nat. Comm., and ready for publication.

Discussion

As I described in my previous review, the experimental work appears to be excellent. There were some flaws in the discussion, but those have been fixed. However, in this case, the results on their own do not constitute a high impact result and the ensuing analysis, including PIC simulations, is critical for assessing suitability for this journal. Broadly speaking, I felt that the authors had either overstated some aspects of their case or did not address complicating factors. The difference between 2D and 3D PIC and the target thickness dependence were two such issues. It was not so much that I thought there were flaws in the analysis as that I couldn't tell if there were. For example, 3D effects can be substantial for these interactions and the lack of 3D results when they said they had them was puzzling, but the new discussion is reassuring. I feel the authors have now addressed my concerns in their response to me and that they have modified the manuscript suitably. I also raised a number of technical concerns (interpretation of the back-reflected spectra, quasi-static B fields, etc) that were also addressed well. I think the paper is now significantly more nuanced and does a better job of clearly describing the uncertainties in their findings.

I read the other reviewers' reports with interest and I am in agreement with them. In particular, Reviewer #2 made excellent points and raised some of the same concerns that I did. One difference is that Reviewer #2 was perhaps less persuaded of the appropriateness of the work for this journal. This may be explained by somewhat different emphases in our reviews. As I explained in my first review, while I also had concerns with VLA as motivation, I focused on the relativistic transparency physics, finding that interesting and sufficient. I believe the revised introduction now does a much better job of motivating this work and placing it in perspective to other work on VLA and on possible applications. This is a significant improvement and removes my last concern.

In summary, I recommend this work for publication.

Re: We greatly appreciate the reviewer for the very helpful comments during the review process.

Reviewer #2 (Remarks to the Author):

Review

I find the authors' responses to the reviewers to be clear, precise and convincing. I understand that the reproduction by numerical simulations of an experiment as complex and characterized by parameters as difficult to control requires adjustments in the input data of the numerical code. It is therefore complicated to unravel the respective influences of the local laser intensity, the accurate thickness of the target or the 3D vs. 2D effects in PIC simulations. However, I believe this does not call into question the main purpose of the paper nor the seriousness of the results obtained. As a consequence, with some minor recommendations (see below), I think this manuscript will be suitable for publication in Nature Communications.

Re: We greatly appreciate the reviewer for the favorable recommendation based on the revised manuscript.

Minor corrections:

(1) The argument regarding the volumetric RT process involved in these experiments is convincing to stand out from other acceleration methods such as the wakefield acceleration. Thus, if the authors believe that VLA has the potential to accelerate micro-Coulomb electron charge bunches, i.e. a charge being about three orders of magnitude higher of what is frequently obtained in wakefield experiments, then it would be useful to state this in the introduction.

Re: Thanks for the suggestion. We have added a sentence (close to the end of 'Introduction') about the potential to accelerate μC electron beams.

(2) The precision brought on the characteristics of the laser plasma interaction is welcome. I guess that the focal spot presented in Fig. S1 has been obtained for a low energy shot (some tens of micro-Joule) and not at full energy? If it is the case, this may be indicated in the figure caption (the intensity profile using full-laser amplification may differ from the one using only the pilot).

Re: Sorry about the lack of clarity. Yes, Fig. S1 was obtained for a low-energy shot and scaled to the full energy. We have now clarified this point in the figure caption.

(3) I have small concerns regarding numerical values: (i) in Methods, the solid angle of the electron spectrometer is stated to $50 \mu\text{sr}$ that does not match the result of $\Delta\Omega = \Delta S/R^2 = 4 \text{ mm}^2/(500 \text{ mm})^2 = 16 \mu\text{sr}$; (ii) in S1, the (mean) laser intensity obtained with 32% of 6.8 J contained in a diameter of $1.7 \mu\text{m}$ and during 75 fs is $1.25 \cdot 10^{21} \text{ W/cm}^2$ (a quantity that is

even further away from the one chosen in the simulations), and not the peak value of $0.9 \times 10^{21} \text{ W/cm}^2$ written in the paper; how is this peak intensity calculated?

Re: Thanks for the corrections.

Yes, the solid angle should be $16 \mu\text{sr}$, not $50 \mu\text{sr}$. We have now corrected this number.

Regarding the peak intensity, we note that the 32% energy is encircled within the FWHM spot area (effective spot) and **full** pulse duration ($\sim 150 \text{ fs}$). Then we estimated the peak intensity using the **FWHM** duration 75 fs as

$$\text{peak intensity} = \frac{6.8 \times 0.32}{\text{effective spot area} \times 75 \text{ fs}}$$

where we also made approximations in calculating the effective spot area. In our measurement, instead of being a circle, the focal spot is more like a rectangle of size $1.81 \mu\text{m} \times 1.73 \mu\text{m}$ due to the limited number of pixels in our CCD image. By plugging in these numbers, we got the peak intensity $\sim 0.9 \times 10^{21} \text{ W/cm}^2$.

We would like to point out that this is a rough estimation. The actual peak intensity at full power was not measured directly. Any imperfections in the focusing optics including, for example, a slight offset in the focal plane position would lead to a reduction in the on-target peak intensity.

We have now added more illustrations on the peak intensity estimation in the Supplementary Information.

(4) I would point out to the authors that the results presented in Ref. 19 (G. Malka et al.) have been the subject of controversies: see the two comments and replies of Kirk T. McDonald, Phys. Rev. Lett. 80, 1350 (1998) and of P. Mora and B. Quesnel, Phys. Rev. Lett. 80, 1351 (1998).

Re: Thanks for pointing this out. We understand there are controversies regarding the theoretical modeling of the experimental results of Ref. 19. Since the work (ref. 19) is mainly about an experimental effort, we think it is appropriate to cite this experiment here as an example to show the difficulty in demonstrating VLA.

Reviewer #3 (Remarks to the Author):

The authors have provided a robust response to both my critique and that of the other referees, now including additional 3D simulations which seem to further corroborate the results. I see no reason to change my previous view that the work should be published.

Re: We greatly appreciate the reviewer for the very helpful comments during the review process.